# From Pixels to Paths: A Multi-Agent Framework for Editable Scientific Illustration

## Abstract

Scientific illustrations demand both high information density and post-editability. However, current generative models have two major limitations: Frist, image generation models output rasterized images lacking semantic structure, making it impossible to access, edit, or rearrange independent visual components in the images. Second, code-based generation methods (TikZ or SVG), although providing element-level control, force users into the cumbersome cycle of "writing-compiling-reviewing" and lack the intuitiveness of manipulation. Neither of these two approaches can well meet the needs for efficiency, intuitiveness, and iterative modification in scientific creation. To bridge this gap, we introduce VisPainter, a multi-agent framework for scientific illustration built upon the model context protocol. VisPainter orchestrates three specialized modules-a Manager, a Designer, and a Toolbox-to collaboratively produce diagrams compatible with standard vector graphics software. This modular, role-based design allows each element to be explicitly represented and manipulated, enabling true element-level control and any element can be added and modified later. To systematically evaluate the quality of scientific illustrations, we introduce VisBench, a benchmark with seven-dimensional evaluation metrics that ensures balanced difficulty across groups and supports streaming updates. It assesses high-information-density scientific illustrations from four aspects: content, layout, visual perception, and interaction cost. To this end, we conducted extensive ablation experiments to verify the rationality of our architecture and the reliability of our evaluation methods. Finally, we evaluated various vision-language models, presenting fair and credible model rankings along with detailed comparisons of their respective capabilities. Additionally, we isolated and quantified the impacts of role division, step control, description quality, and reference images on the quality of illustrations.

## 1 Introduction

Scientific diagrams, including model schematics, experimental workflows, and graphical abstracts, are central to reporting methods and results. Unlike natural images, these figures must encode explicit structure such as modules, connectors, and labels, maintain precise geometry, and remain editable throughout the publication lifecycle. Vector formats are therefore the practical default in authoring and revision. At the same time, modern text-to-image diffusion has made rapid progress in photo realistic synthesis (Jain et al., 2023; Rombach et al., 2022; Saharia et al., 2022), and conditioning by grounding or auxiliary signals has improved spatial fidelity (Li et al., 2023; Zhang et al., 2023b; Mou et al., 2023; Xie et al., 2023). However, the outputs of these systems are almost always raster images without element semantics. Arrows, annotations, and blocks cannot be individually accessed or modified after generation. This gap limits the use of recent generative advances for scientific illustration, where editability and structure are as important as visual realism.

Prior work covers parts of this need but does not provide an end-to-end, interactive solution. Code-based generation methods (TikZ, SVG, etc.) (Belouadi et al., 2024a;b; Carlier et al., 2020; Wu et al., 2025; Jain et al., 2023; Yang et al., 2025b) provide element-level control. However, this code-compilation-based approach is different from operating on a canvas; it forces users into a cumbersome cycle of writing, compiling, and reviewing, which is not conducive to the rapid iteration of scientific diagrams. On evaluation, recent benchmarks assess scientific image generation on correctness (Zhang et al., 2024) or chart-to-code capabilities (Yang et al., 2025a). These benchmarks target

relatively simple scientific diagrams or data visualization chart content, and do not involve high-information-density schematic diagrams, conceptual diagrams, etc. Other works focus on parsing and question answering (Kembhavi et al., 2016; Methani et al., 2020; Tannert et al., 2022; Mathew et al., 2020; Singh et al., 2024), rather than the quality of diagram generation. Therefore, there is no standard way to translate free-form instructions into editable vector diagrams and to measure not just what is created, but how efficiently it meets the functional demands of scientific communication.

We address these gaps with **VisPainter** and **VisBench**. VisPainter is a multi-agent framework built on the Model Context Protocol (MCP) (Hou et al., 2025). It enables the creation of scientific illustrations by operating professional software through over 30 integrated MCP server tools. This allows each visual element to be represented and manipulated, thereby achieving element-level control and iterative optimization. (Li et al., 2023; Zhang et al., 2023b) VisBench is the first benchmark for evaluating high-information-density diagrams such as schematic diagrams and conceptual diagrams in research papers. It measures the quality of diagrams through seven metrics across four dimensions: accuracy, recall, design error, blank space, readability, alignment, and interaction steps. Through extensive evaluations and ablation experiments on various vision-language models, we have demonstrated the stability and credibility of the framework design and evaluation metrics, and presented fair model comparisons, fine-grained analyses of various capability values, as well as the impact of different design factors on capabilities. (Mou et al., 2023; Xie et al., 2023; Saharia et al., 2022).

**Contributions.** (1) **VisPainter**: A multi-agent framework that turns natural language and mixed-modality instructions into editable vector diagrams with explicit element handles; (2) **VisBench**: A benchmark for scientific illustration with a seven-dimensional evaluation protocol targeting content, layout, readability, and interaction cost; (3) **Comprehensive evaluation**: Extensive model comparisons and ablation experiments verify the benefits of our framework, demonstrate the reliability of our metrics, and present fair model comparisons and detailed capability analyses.

## 2 RELATED WORK

**Image Generation Models** Recent progress in generative models, particularly those based on Transformers, has revolutionized text-to-image synthesis (Vaswani et al., 2023; Ramesh et al., 2021; 2022; Betker et al., 2023; Rombach et al., 2022). To improve spatial fidelity, some methods incorporate auxiliary controls, such as bounding boxes, through models like GLIGEN (Li et al., 2023) and ControlNet (Zhang et al., 2023b), while others leverage compositional planning (Feng et al., 2023; Bao et al., 2023; Zhang et al., 2023a; Xie et al., 2023; He et al., 2023; Xiao et al., 2024). However, these approaches primarily generate raster images, lacking the element-level structure required for post-generation editing. This makes them unsuitable for scientific diagrams where manipulation of individual components is essential. While recent vector-native models like DeepSVG (Carlier et al., 2020), Im2Vec (Reddy et al., 2021), and VectorFusion (Jain et al., 2023) show promise, they mainly focus on simpler graphics like icons, not dense, structured scientific diagrams.

**Code-Based Generation Approaches** An alternative approach emits diagram code to achieve structural control. This spans from general-purpose formats like TikZ and SVG, targeted by systems such as AutomaTikZ (Belouadi et al., 2024a;b) and StarVector (Rodriguez et al., 2024; Wu et al., 2025), to more constrained domain-specific languages (DSLs) like PlantUML (Roques, 2009) and Mermaid (Sveidqvist & contributors, 2014). Empirical studies (Zou et al., 2024) suggest that large language models are more adept at handling structured languages such as TikZ than low-level SVG paths. Despite offering high-level semantic control, these methods all depend on a non-interactive write-compile-review loop. This process is cumbersome and ill-suited for the iterative refinement required for complex scientific schematics. Instead of generating code, VisPainter facilitates atomic GUI operations (e.g., insert, align) within a vector editor. This enables real-time manipulation without DSL expertise and allows users to directly participate in the drawing process.

**Scientific Diagram Benchmarking and Evaluation** Existing benchmarks primarily assess aesthetics on static images (Chen et al., 2015; Saharia et al., 2022) or their compositional correctness (Huang et al., 2025; Zhang et al., 2024). Specialized benchmarks often bypass open-ended generation, focusing instead on tasks like chart-to-code (Yang et al., 2025a) or diagram understanding (Kembhavi et al., 2016; Methani et al., 2020; Tannert et al., 2022; Singh et al., 2024). While some resources do target scientific content (Chang et al., 2025; Wang et al., 2025), they do not evaluate the generation of structured, vector-editable outputs. Even vector-focused frameworks like VGBench (Zou et al., 2024) and SVGEditBench (Nishina & Matsui, 2024) evaluate code-level ac-

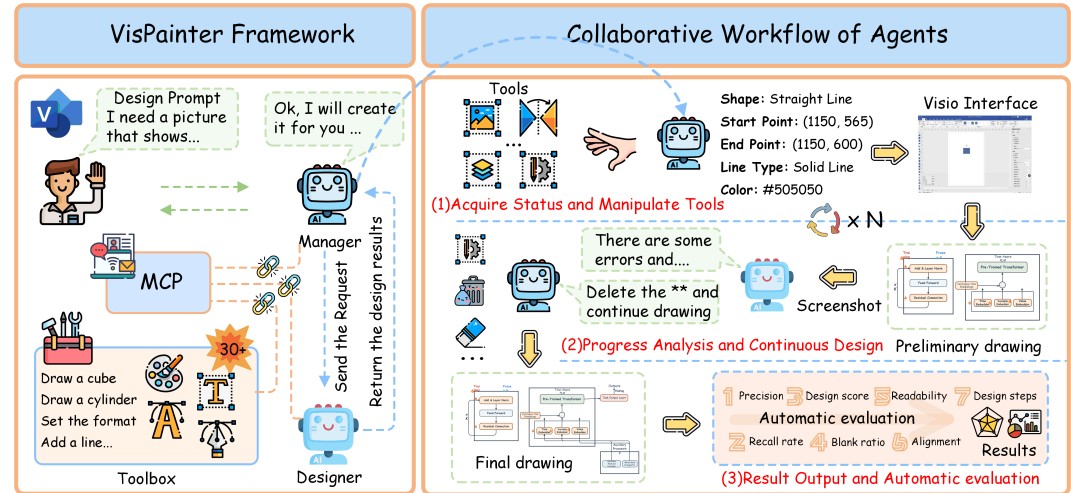

Figure 1: VisPainter framework diagram and workflow diagram.The overall workflow proceeds as follows: (1) the Manager parses the user request into a structured task and locates the relevant Visio functions; (2) the Designer generates an initial draft layout; (3) the Manager invokes Toolbox operations to render the draft and captures a screenshot; (4) the Designer iteratively updates the layout based on feedback until convergence. The final output includes both a bitmap preview for quick inspection and a vector source file that can be further modified within Visio or other editors.

curacy, overlooking the GUI-level interaction cost. Consequently, a standard is needed to assess the generation of high-information-density schematics, considering both their final quality and the efficiency of the interactive process.

## 3 METHOD

### 3.1 FRAMEWORK

VisPainter is a multi-agent framework designed to translate free-form instructions into editable scientific diagrams. Current generative models typically produce static, rasterized images or vector graphics compiled by code, which decouples the user from the creation process. VisPainter addresses this by directly orchestrating a standard vector graphics editor, making the generation process both observable and interactive. It operates on the MCP (Model Context Protocol) service stack provided by `cline`, an open protocol enabling language models to call external tools (see App.B). This multi-agent decomposition is critical: it separates high-level reasoning from low-level GUI operations, mirroring the cognitive division of labor in human-computer interaction. We instantiate this philosophy in a Manager-Designer-Toolbox architecture, encapsulating over thirty Visio functions as callable MCP tools. The workflow is detailed in Fig. 1.

**Manager:** The Manager interprets human instructions or intermediate drafts produced by the Designer, maintains the task state, and dispatches appropriate tool calls. Each request is wrapped as an MCP message, executed in the Toolbox, and logged with full version control, design steps, and token usage for later rollback and auditing. By keeping a consistent record of interactions, the Manager ensures that the design process remains reproducible and transparent.

**Designer:** The Designer, which can be instantiated with any capable Vision-Language Model (VLM), is the creative engine. Given a directive from the Manager, it generates a structured draft specifying shapes, labels, and their spatial relationships. After each drawing cycle, the Designer receives the rendered screenshot and refines the layout. This iterative loop emulates the essential human design process of drafting and refinement, allowing the system to progressively correct errors and improve visual coherence.

**Toolbox:** The Toolbox serves as the execution layer, wrapping low-level GUI/COM operations of Visio into a set of stateless MCP tools (e.g., shape insertion, alignment). Each tool call produces a directly editable vector object and returns a corresponding screenshot for the Designer's feedback loop. By abstracting away the complexities of the GUI, the Toolbox allows the Designer and Manager to focus on semantic and layout decisions, rather than implementation details.

## 3.2 BENCHMARK

The VisPainter framework provides the capability to automatically generate editable diagrams, but this new capability creates a critical gap. While benchmarks for scientific figures exist(Zhang et al., 2024; Yang et al., 2025a), they focus on factual correctness in plots or chart-to-code translation, not the generation of editable, high-information-density schematics like workflows and model architectures. To fill this void, we introduce VisBench, the first streaming benchmark specifically designed for evaluating the generation of such structured, vector-based diagrams. VisBench not only evaluates models on content integrity, layout quality, and perceptual correctness, but crucially, it also measures interaction cost and integrates these dimensions into a unified score (see Section 3.2.2). This benchmark supports two settings: text-to-image (T2I) and text + image-to-image (TI2I).

### 3.2.1 BENCHMARK CONSTRUCTION

**Data Collection and Annotation.** We curated a corpus of 360 high-information-density scientific diagrams from recent open-access publications (2022-2025) across diverse STEM fields. The collection process involved a pipeline: (1) harvesting thousands of vector-based illustrations from sources like arXiv and GitHub; (2) automatic pre-filtering with GPT-4o to remove unsuitable images; (3) three rounds of manual checks by experts to ensure high complexity and drawability. Each selected diagram was then annotated with a three-layer description (layout, text, visual elements) via a human-in-the-loop process. While we acknowledge that annotation quality could be further improved, our primary goal is to establish a stable testbed for relative model comparison. Our ablation studies confirm that model rankings are robust even with varied annotation quality. A detailed description of the data collection pipeline, licensing, and corpus statistics is provided in Appendix K.

**Difficulty definition and balanced sampling.** A key challenge in benchmarking is ensuring fair comparison across different evaluation runs. A common approach is to stratify the dataset by domain or predefined difficulty levels . However, we found these methods problematic for scientific diagrams: domain labels often confound subject matter with visual style, and manually assigning discrete difficulty levels is subjective and fails to capture the continuous nature of complexity. To address this, we adopt a more robust, data-driven approach. We define difficulty objectively using a continuous metric-the number of graphical components (`element_count`)-and aim to make each monthly evaluation cohort a statistically representative microcosm of the entire benchmark. The formal objective is to find a subset $S$ that minimizes:

$$J(S) = \left| \mu_S - \mu \right| + \lambda \left| \sigma_S - \sigma \right|. \tag{1}$$

This objective function penalizes deviations in the cohort's mean ($\mu_S$) and standard deviation ($\sigma_S$) from the global dataset statistics ($\mu$, $\sigma$), with $\lambda$ weighting the latter. Our two stage process first guarantees broad coverage by sampling from $L$ difficulty quantiles. A lightweight Metropolis Hastings refinement then iteratively swaps elements to optimize $J(S)$ until a convergence threshold is met. This procedure yields cohorts with stable difficulty, thereby insulating model comparisons from sampling bias. The hyper-parameters ($L, \lambda$) and a full validation of the method's robustness are detailed in the Appendix F.

**Dynamic update policy.** To combat benchmark stagnation and model overfitting, VisBench employs a "seasonal" update schedule. The current corpus contains 360 diagrams. For the current evaluation season, we pre-sample 12 monthly cohorts (360 diagrams in total) to ensure stationary difficulty. Concurrently, we curate new data. Every four months, 120 newly collected and annotated diagrams are added to a staging pool. These new items are frozen and will only be used for sampling in the next evaluation season. This streaming approach ensures the benchmark remains relevant against rapid model development while maintaining the statistical integrity of the current season's evaluations.

### 3.2.2 EVALUATION METRICS

VisBench is designed to assess whether models can generate scientific schematics that are (accurate in content, well-aligned in layout, readable in text, and efficient to produce with minimal trial-and-error). To operationalize this goal, we evaluate four aspects: content fidelity, layout quality, visual perception and interaction cost. All scores are computed from the exported vector PDF with intact structure; parsing is object-level unless otherwise stated.

**Content level.** Precision and Recall measure whether required textual elements are correctly rendered in the output. The requirement set $P$ is derived from task metadata; the generated set $G$ is extracted from the PDF's structured text objects. Before matching we normalize all text by stripping punctuation and lowercasing. We then compute

$$\text{Precision} = \frac{|P \cap G|}{|G|}, \qquad \text{Recall} = \frac{|P \cap G|}{|P|}. \tag{2}$$

**Layout level.** The Blank-space score measures canvas utilization. We rasterize the output to a normalized size (1024px on the short side), overlay a 128px grid, and use a vision model to estimate the blank ratio $\beta$. The score is then mapped via:

$$\text{Blank} = \frac{1}{1 + 2\beta}. \tag{3}$$

The 128 px grid choice and a human comparison are justified by ablations in the Appendix E. Alignment score measures row and column regularity. Let $g(x, y)$ be the grayscale at pixel $(x, y)$ and $W, H$ the raster width and height. We compute the vertical projection $p(y) = \frac{1}{W} \sum_{x=1}^{W} g(x, y)$ and define the score as:

$$\text{Align} = \frac{1}{1 + \text{Var}_y[p(y)]/10^4}. \tag{4}$$

**Perceptual level.** Design-error score counts visually evident mistakes in lines, connectors, modules, overlaps, and spillover. We rasterize the figure and ask GPT-o3 to identify discrete errors; with error count $e$ the score is

$$\text{Design} = \frac{1}{1 + 2e}. \tag{5}$$

We ablate the vision judge in the Appendix D to show robustness across alternatives. Readability quantifies how much of the intended text remains legible after rendering, accounting for occlusion, small fonts, and low contrast. Let $R$ be the set of normalized strings that are visually readable in the rasterized output; the score is

$$\text{Readability} = \frac{|R \cap G|}{|G|}. \tag{6}$$

This metric penalizes cases where text exists in the vector file but is not practically visible.

**Overall score.** We aggregate the six metrics (Precision, Recall, Design, Blank, Readability, Align), each defined on $[0, 1]$, with fixed weights $(0.20, 0.20, 0.20, 0.05, 0.25, 0.10)$ to obtain a base score $s$. (Weighting details and a sensitivity analysis with equal weights are reported in the Appendix C.) Interaction cost is measured by the step count $n$ of valid atomic drawing commands recorded in the MCP trace. Dynamic hyper-parameters are fixed for each evaluation season:

$$K = \text{mean}_i(n_i), \qquad r = 1 - \text{mean}_i(s_i).$$

We combine $s$ and $n$ in the following way to obtain the Dynamic Quality Score (DQS):

$$\text{DQS}(s, n) \;=\; s\left(1 - (1 - s)\,\text{sat}(n)\right) \;+\; r\,s\,(1 - \text{sat}(n)), \quad \text{with } \text{sat}(n) = \frac{n}{n + K}. \tag{7}$$

**Why DQS?** Using $s$ alone ignores efficiency: two systems with similar quality may differ greatly in how many steps they take. DQS integrates quality and cost. When $n < K$, $\text{sat}(n)$ is small and DQS rewards concise traces; when $n$ grows, the penalty becomes stronger and DQS declines, discouraging long trial-and-error loops. The adaptive factor $r = 1 - \text{mean}_i(s_i)$ ties tolerance to cohort difficulty: on easier cohorts (higher mean $s$) extra steps are penalized more, while on harder cohorts DQS is more forgiving. This yields a fairer "quality-per-cost" measure across systems.

## 4 EXPERIMENTS

### 4.1 EXPERIMENTAL SETUP

**Dataset:** We adopt the difficulty-balanced sampling protocol to form fixed monthly cohorts of 30 diagrams (15 T2I, 15 TI2I) from VisBench. A season pre-commits 12 such cohorts under a fixed seed; new data are staged for the next season. This paper reports results on the first three monthly cohorts

already completed (90 diagrams in total), which jointly test from-scratch (T2I) and reference-guided (TI2I) diagram creation.

**Evaluation:** The evaluation uses the seven-dimensional metrics proposed in the previous text to calculate the weighted score. The weighted base score is further processed by the step-penalty/bonus function, yielding an overall quality score DQS. All drawing results are PDF files exported with structured content, and all scoring scripts run offline on the same workstation. To ensure reproducibility, API responses are cached.

**Models:** We compare nine publicly accessible vision-language models: GPT-4o, GPT-4.1, GPT-o3, GPT-5, Claude-Opus-4, Gemini-2.5-Pro, Qwen-VL-Max, Qwen2.5-VL-72B, and Llama-4-Maverick. Each system is queried via its official REST API (versions no earlier than 04-14-2025). Unless stated otherwise, temperature is 0.2, the per-round design stride is 1, and other parameters remain at vendor defaults. A unified prompt template is used across models; further details and endpoints are provided in the Appendix J.

## 4.2 QUANTITATIVE RESULTS

Our main results, summarized in Table 1, reveal distinct trade-offs across models in content completeness, generation efficiency, and layout quality. A visual breakdown of these capabilities is provided in Fig. 3. **In the T2I setting**, Gemini-2.5-Pro and GPT-5 achieve the highest DQS scores by combining strong recall with stable readability and blank-space control, though both rely on long interaction traces. GPT-o3 excels at precision and text rendering, reflecting strong parsing ability, but it tends to miss less salient elements. Claude-Opus-4 and GPT-4.1 reach balanced scores, yet their reliance on repeated self-corrections drives up step counts. Models such as Qwen-VL-Max, Qwen2.5-VL-72B, and Llama-4-Maverick score lowest, mainly due to unstable design fidelity and weaker recall on dense diagrams. **In the TI2I setting**, recall improves across all systems because of reference-image guidance, but the mean step count rises substantially, penalizing models that already depend on long editing sequences. Gemini-2.5-Pro and GPT-5 again lead, with GPT-5 slightly outperforming Gemini on overall balance, though at high computational cost. GPT-o3 maintains clean readability but continues to trade coverage for precision. The lighter systems remain fragile under layout constraints, with frequent element misplacement that depresses design scores.

## 4.3 QUALITATIVE STUDY

We present representative output results in Fig. 2, which helps explain the numerical gaps. GPT-4o produces results quickly with minimal steps, but parts of diagrams often fall off-canvas or are omitted, limiting recall. GPT-4.1 benefits from undo and redo operations that fix local mistakes, yet inaccurate font-size estimates introduce new errors and inflate steps. GPT-o3 follows instructions closely and renders text cleanly, yielding strong precision and readability, but it sometimes overlooks smaller components. Claude-Opus-4 delivers consistently clean, well-aligned layouts, though frequent micro-adjustments increase interaction cost without proportional gains. Gemini-2.5-Pro generates content-rich diagrams and handles complex prompts, but this breadth occasionally reduces precision when details are over-interpreted. GPT-5 shows stronger global planning and more stable typography than Gemini, especially in TI2I: it integrates the reference image to improve spacing and hierarchy, reducing off-canvas placement and overlaps. Its main drawback is longer self-check loops, which raise step counts; in some dense cases, conservative regularization sacrifices recall. Qwen-VL-Max and Qwen2.5-VL-72B show unstable element placement, with off-canvas or weak alignment lowering design scores. Llama-4-Maverick lacks a clear stopping strategy; redundant strokes and repetitive edits waste steps and often break formatting on complex graphs.

## 5 ABLATION STUDY

To verify the rationality and reliability of the VisPainter and VisBench, We conducted ablation experiments on its core factors: **role configuration**, **step granularity**, and **description quality**. For more ablation experiments, see Appendix C, D, E. These aim to clarify the impact of each component on image drawing, thereby explaining why a complete system design is necessary and why the evaluation metrics are reliable. All experiments were conducted on the same tasks (15 T2I and 15 TI2I). We report the average score of each model. We adopted a lightweight setup (30 tasks) because the purpose of the experiment is to analyze trends rather than establish leaderboard results. The evaluation metrics follow Section 3.2.2, and representative outputs are provided in the Appendix.

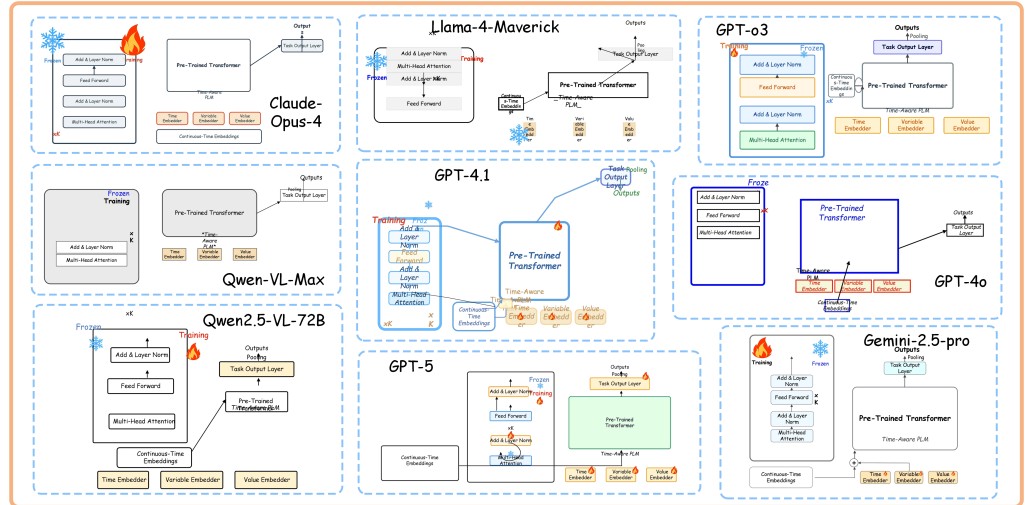

Figure 2: Examples of scientific plotting outputs from different models.

Table 1: Model evaluation experiment results: Performance of each model in T2I (upper part) and TI2I (lower part) scenarios. Higher is better except "Steps".

| Model | Precision | Recall | Design | Blank | Read. | Align. | Steps(↓) | Score(s) | DQS |
|---|---|---|---|---|---|---|---|---|---|
| Gemini-2.5-Pro | 0.92 | 0.88 | 0.53 | 0.84 | **0.89** | 0.91 | 29.83 | **0.82** | **0.85** |
| GPT-5 | 0.89 | 0.83 | **0.56** | **0.88** | 0.88 | 0.90 | 26.90 | 0.81 | 0.84 |
| GPT-o3 | 0.87 | 0.78 | 0.52 | 0.79 | 0.88 | 0.93 | 23.43 | 0.79 | 0.82 |
| Claude-Opus-4 | 0.89 | **0.88** | 0.44 | 0.86 | 0.82 | 0.93 | 33.91 | 0.78 | 0.78 |
| GPT-4.1 | 0.84 | 0.80 | 0.41 | 0.80 | 0.67 | **0.93** | 27.05 | 0.71 | 0.70 |
| GPT-4o | **0.95** | 0.52 | 0.37 | 0.72 | 0.72 | 0.89 | **19.51** | 0.67 | 0.68 |
| Qwen2.5-VL-72B | 0.78 | 0.73 | 0.40 | 0.74 | 0.72 | 0.85 | 26.44 | 0.69 | 0.67 |
| Llama-4-Maverick | 0.82 | 0.67 | 0.37 | 0.80 | 0.79 | 0.83 | 30.71 | 0.69 | 0.67 |
| Qwen-VL-Max | 0.76 | 0.75 | 0.41 | 0.68 | 0.69 | 0.90 | 27.79 | 0.68 | 0.66 |
| GPT-5 | 0.77 | 0.70 | **0.56** | 0.86 | **0.84** | 0.92 | 44.19 | **0.75** | **0.77** |
| Gemini-2.5-Pro | 0.81 | **0.72** | 0.50 | 0.87 | 0.81 | 0.92 | 46.72 | 0.74 | 0.75 |
| GPT-o3 | 0.73 | 0.65 | 0.46 | 0.85 | 0.78 | 0.93 | 32.26 | 0.70 | 0.73 |
| Claude-Opus-4 | 0.75 | 0.63 | 0.47 | **0.88** | 0.83 | **0.93** | 47.85 | 0.71 | 0.71 |
| GPT-4o | **0.84** | 0.44 | 0.40 | 0.82 | 0.73 | 0.90 | **29.88** | 0.65 | 0.67 |
| GPT-4.1 | 0.73 | 0.59 | 0.42 | 0.79 | 0.65 | 0.90 | 31.31 | 0.64 | 0.65 |
| Qwen-VL-Max | 0.73 | 0.41 | 0.39 | 0.79 | 0.81 | 0.87 | 36.83 | 0.64 | 0.63 |
| Qwen2.5-VL-72B | 0.68 | 0.45 | 0.35 | 0.82 | 0.72 | 0.88 | 36.30 | 0.61 | 0.59 |
| Llama-4-Maverick | 0.62 | 0.47 | 0.31 | 0.81 | 0.74 | 0.86 | 37.71 | 0.59 | 0.57 |

## 5.1 ROLE CONFIGURATION

To verify the necessity of VisPainter's explicit division of labor, we create a baseline by merging the Manager and Designer into a single agent responsible for both high-level planning and low-level tool execution. We evaluate this unified agent using GPT-4o and Gemini-2.5-Pro, keeping all other experimental conditions identical. The results are summarized in Table 2.

**Findings:** (1) Content Fidelity Collapses: Merging roles causes a catastrophic drop in performance. The overall score (s) falls by an average of 0.14, driven by a sharp decline in Precision (-23% avg.) and Recall (-27% avg.). This indicates that without a dedicated Manager for global planning, the unified agent systematically omits required elements. (2) Layout Coherence Degrades: Both Design (-0.13 avg.) and Alignment (-0.05 avg.) scores also decrease, confirming that the separation of concerns is vital for maintaining structural integrity. (3) Anomalous Readability Increase: Interestingly, Readability and Blank-space scores sometimes show a slight increase. We attribute this to the lower element count; with fewer objects to draw, the model can afford larger fonts and more generous spacing, paradoxically improving the legibility of the remaining content. In summary, these results indicate that in the design of the VisPainter architecture, it is crucial to separate high-level planning from low-level execution in order to simultaneously achieve content integrity and layout quality.

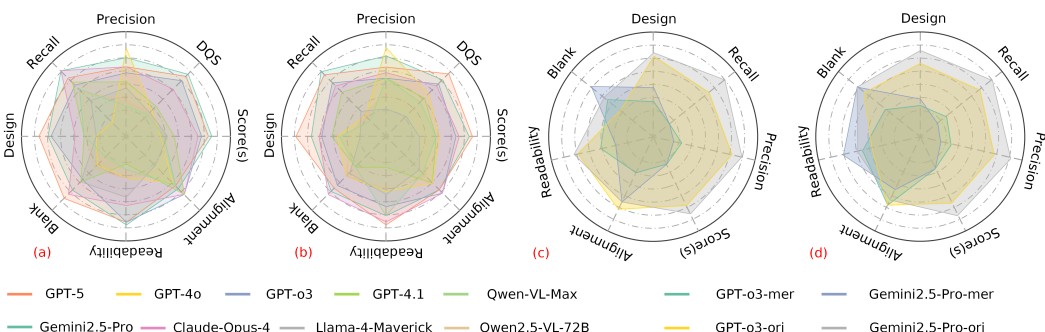

GPT-5   GPT-4o   GPT-o3   GPT-4.1   Qwen-VL-Max   GPT-o3-mer   Gemini2.5-Pro-mer

Gemini2.5-Pro   Claude-Opus-4   Llama-4-Maverick   Qwen2.5-VL-72B   GPT-o3-ori   Gemini2.5-Pro-ori

Figure 3: Comparison of 9 model capabilities: T2I test scenario (a); TI2I test scenario (b). Comparison of model capabilities before and after role configuration ablation experiments: T2I test scenario (c); TI2I test scenario (d). "mer" stands for role integration, and "ori" stands for baseline setting.

Table 2: Results of the role configuration ablation experiment. Performance tested in T2I (upper part) and TI2I (lower part) scenarios. The differences are relative to the full model in Table 1.

| Model | Precision | Recall | Design | Blank | Readability | Alignment | Score(s) |
|---|---|---|---|---|---|---|---|
| GPT-o3 | 0.62-0.15 | 0.53-0.25 | 0.39-0.13 | 0.83+0.04 | 0.82-0.06 | 0.83-0.10 | 0.64-0.15 |
| Gemini-2.5-Pro | 0.58-0.34 | 0.53-0.35 | 0.43-0.10 | 0.88+0.04 | 0.78-0.12 | 0.91+0.00 | 0.64-0.18 |
| GPT-o3 | 0.52-0.21 | 0.46-0.19 | 0.33-0.13 | 0.79-0.06 | 0.80+0.02 | 0.93+0.00 | 0.59-0.11 |
| Gemini-2.5-Pro | 0.47-0.33 | 0.41-0.31 | 0.35-0.15 | 0.87+0.00 | 0.84+0.03 | 0.90-0.02 | 0.59-0.15 |

## 5.2 STEP GRANULARITY

This experiment probes the trade-off between interaction efficiency and generation quality by varying the step granularity. We control the maximum number of elements the Designer can generate per turn, testing values of $n \in \{1, 2, 4, 8, 16\}$, while holding the model configuration (GPT-4o Manager, Gemini-2.5-Pro Designer) and other settings constant. The results are summarized in Table 3.

**Findings:** Results reveal a trade-off between efficiency and quality. (1) Optimal Range ($n \leq 4$): Increasing granularity from $n = 1$ to $n = 4$ reduces interaction steps (from 38.2 to 10.5 avg.) with no loss in quality score (s). This range is the sweet spot, where batching operations improves efficiency without compromising quality. (2) Degradation and Collapse ($n \geq 8$): At $n = 8$, performance degrades with emergent layout errors. At $n = 16$, the system collapses, showing sharp drops in Precision (avg. -0.26) and Design (avg. -0.18). This indicates larger step sizes exceed model's single-turn planning capacity, leading to "cognitive congestion" and catastrophic layout failures. In conclusion, a small step size ($n \leq 4$) offers the best balance. Larger values push the model beyond its planning limits, causing content and layout errors.

## 5.3 IMPACT OF DESCRIPTION QUALITY

This experiment tests the benchmark's robustness to variations in prompt quality. We create a "low-quality" variant by stripping detailed attributes (e.g., colors, positions, sizes) from the original prompts, reducing their average length by 28% (from 434.7 to 313.9 words). We then evaluate 7 models on both the original and degraded descriptions. Results are reported in Table 4 and Fig. 4.

**Findings:** (1) Ranking Stability: The most critical finding is that relative model rankings remain stable. While absolute scores fluctuate with prompt quality, the performance gaps between models are preserved. This validates VisBench's ability to measure fundamental capability differences, which is our primary goal. (2) Counterintuitive Fidelity: Simplified descriptions paradoxically improve Precision and Recall. With fewer detailed constraints, models are less likely to omit or misinterpret core requirements, leading to better content fulfillment. (3) Modality Trade-off: The impact of sparse text depends on the setting. In T2I, lacking details predictably harms Design scores. In TI2I, however, the reference image largely compensates for this loss. This suggests that when a strong visual prior is present, overly verbose text can become redundant or even conflicting. These findings confirm that while description quality affects absolute scores, our benchmark's conclusions about relative model capabilities are robust. This supports our focus on evaluating models' intrinsic abilities rather than their performance on a perfectly annotated dataset.

Table 3: Results of the design step size ablation experiment: Impact of step size settings on T2I (upper part) and TI2I (lower part) scenarios.The differences are relative to the full model in Table 1.

| Step $n$ | Precision | Recall | Design | Blank | Readability | Alignment | Steps(↓) | Score(s) |
|---|---|---|---|---|---|---|---|---|
| 1 | 0.92 | 0.88 | 0.53 | 0.84 | 0.89 | 0.91 | 29.73 | 0.82 |
| 2 | 0.91-0.01 | 0.91+0.03 | 0.49-0.04 | 0.83-0.01 | 0.90+0.01 | 0.91+0.00 | 12.90-16.83 | 0.83+0.01 |
| 4 | 0.92+0.0 | 0.91+0.03 | 0.51-0.02 | 0.86+0.02 | 0.90+0.01 | 0.92+0.01 | 7.35-22.38 | 0.83+0.01 |
| 8 | 0.90-0.02 | 0.87-0.01 | 0.42-0.11 | 0.88+0.04 | 0.91+0.02 | 0.89-0.02 | 5.20-24.53 | 0.80-0.02 |
| 16 | 0.62-0.30 | 0.73-0.15 | 0.33-0.20 | 0.82-0.02 | 0.88-0.01 | 0.89-0.02 | 3.27-26.46 | 0.68-0.14 |
| 1 | 0.81 | 0.72 | 0.50 | 0.87 | 0.81 | 0.92 | 46.72 | 0.74 |
| 2 | 0.83+0.02 | 0.73+0.01 | 0.49-0.01 | 0.83-0.04 | 0.83+0.02 | 0.91-0.01 | 22.10-24.52 | 0.75+0.01 |
| 4 | 0.83+0.02 | 0.72+0.00 | 0.51+0.01 | 0.86-0.01 | 0.83+0.02 | 0.92+0.00 | 13.73-32.89 | 0.75+0.01 |
| 8 | 0.71-0.10 | 0.73+0.01 | 0.46-0.04 | 0.88+0.01 | 0.83+0.02 | 0.89-0.03 | 6.41-40.21 | 0.72-0.03 |
| 16 | 0.59-0.22 | 0.47-0.25 | 0.34-0.16 | 0.88+0.01 | 0.80-0.01 | 0.90-0.02 | 3.88-42.74 | 0.61-0.13 |

Table 4: Ablation experiment on description quality: The impact of low-quality descriptions, T2I (upper part) and TI2I (lower part) scenarios.The differences are relative to the full model in Table 1.

| Model | Precision | Recall | Design | Blank | Readability | Alignment | Score(s) |
|---|---|---|---|---|---|---|---|
| Gemini-2.5-Pro | 0.93+0.01 | 0.89+0.01 | 0.40-0.13 | 0.83-0.01 | 0.90+0.01 | 0.94+0.03 | 0.80-0.02 |
| Claude-Opus-4 | 0.91+0.02 | 0.90+0.02 | 0.37-0.07 | 0.86+0.00 | 0.75-0.07 | 0.94+0.01 | 0.76-0.02 |
| GPT-o3 | 0.88+0.01 | 0.80+0.02 | 0.41-0.11 | 0.88+0.09 | 0.83-0.05 | 0.90-0.03 | 0.76-0.03 |
| GPT-4o | 0.91+0.04 | 0.68+0.13 | 0.35-0.02 | 0.65-0.07 | 0.80+0.08 | 0.93+0.04 | 0.71+0.03 |
| GPT-4.1 | 0.87+0.03 | 0.80+0.00 | 0.36-0.05 | 0.78-0.02 | 0.67+0.00 | 0.92-0.01 | 0.70+0.00 |
| Qwen-VL-Max | 0.84+0.08 | 0.67-0.08 | 0.40-0.01 | 0.72+0.04 | 0.75+0.06 | 0.93+0.03 | 0.69+0.01 |
| Llama-4-Maverick | 0.82+0.00 | 0.72+0.05 | 0.31-0.06 | 0.80+0.00 | 0.77-0.02 | 0.92+0.09 | 0.69+0.0 |
| Gemini-2.5-Pro | 0.82+0.01 | 0.73+0.01 | 0.60+0.10 | 0.82-0.05 | 0.81+0.00 | 0.91-0.01 | 0.76+0.02 |
| Claude-Opus-4 | 0.81+0.06 | 0.69+0.06 | 0.55+0.08 | 0.84-0.04 | 0.77-0.06 | 0.93+0.00 | 0.74+0.03 |
| GPT-o3 | 0.81+0.08 | 0.64-0.01 | 0.52+0.06 | 0.87+0.02 | 0.75-0.03 | 0.90-0.03 | 0.72+0.02 |
| GPT-4o | 0.83-0.01 | 0.47+0.03 | 0.51+0.11 | 0.79-0.03 | 0.77+0.04 | 0.88-0.02 | 0.68+0.03 |
| GPT-4.1 | 0.77+0.04 | 0.61+0.02 | 0.53+0.11 | 0.81+0.02 | 0.67+0.02 | 0.90+0.00 | 0.68+0.04 |
| Qwen-VL-Max | 0.78+0.05 | 0.40-0.01 | 0.49+0.10 | 0.83+0.04 | 0.80-0.01 | 0.88+0.01 | 0.66+0.02 |
| Llama-4-Maverick | 0.66+0.04 | 0.44-0.03 | 0.40+0.09 | 0.83+0.02 | 0.76+0.02 | 0.89+0.03 | 0.62+0.03 |

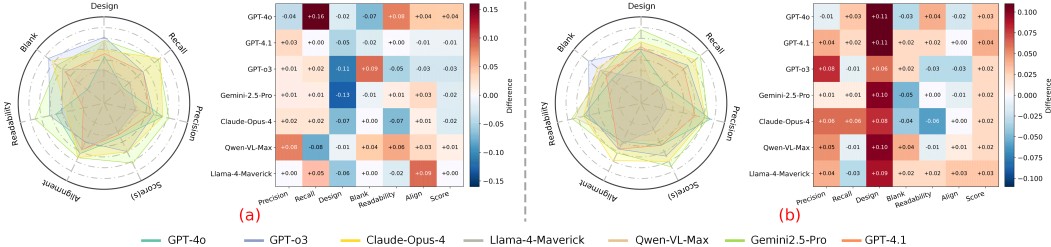

Figure 4: Visualization charts of the ablation experiments for description quality: T2I test scenario (a); TI2I test scenario (b). The radar chart represents the capability comparison between different models in this test, and the heatmap represents the changes in model capabilities compared to the standard description quality.

## 6 CONCLUSION

We introduced VisPainter, a multi-agent framework that generates fully editable scientific diagrams by controlling standard vector graphics software. This approach bridges the gap between generative models and the practical needs of scientific creation. To systematically measure progress, we have developed VisBench, the first streaming benchmark in this field for high-information-density scientific diagrams, which uses a seven-dimensional metric to conduct a comprehensive evaluation from four dimensions: content, layout, visual appearance, and interaction cost. Our extensive evaluations and ablation experiments confirm the rationality of our architecture and the reliability of our evaluation method, and provide fair and credible capability rankings and fine-grained analyses of various indicators for multiple VLMs. At the same time, we also discuss in detail the impact of various factors on model capabilities. We believe VisPainter and VisBench lay the foundation for a new generation of tools that will assist and collaborate with humans in creating structured visual content.

**Reproducibility Statement** The models, prompts, data generation code, and experimental evaluation code we used are all open-source. To ensure the reproducibility of this paper, we have made efforts in the following aspects: (1) The code and data will be open-sourced once the paper is accepted. (2) We have conducted extensive experiments under different settings to verify the general applicability of the proposed framework. (3) We have provided a framework and evaluation methods based on open-source models, significantly improving reproducibility. (4) The multi-agent drawing framework used in this paper, as well as the corresponding MCP Server tool, will be open-sourced simultaneously along with the code and data of this paper.

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

# 7 APPENDIX

## A STATEMENT ON LLMS USAGE

The authors used large language models (LLMs) during the writing process solely for language refinement and editing. It should be explicitly stated that LLMs were not employed in any core aspects of the research, including the formulation of research ideas, the design of methodologies, the execution of experiments, or the development of conclusions. All scholarly contributions were made independently by the authors.

## B SUPPLEMENTARY NOTES ON THE EXPERIMENTAL PLATFORM

**MCP Overview** The Model Context Protocol (MCP) is an open standard that specifies how language models can interact with external tools, services, and data sources through a unified interface. By abstracting tool calls into standardized messages, MCP enables models to request operations, retrieve contextual resources, and integrate third-party functions without custom engineering for each service. A server exposes tools and resources, while a client (such as a model or orchestrator) issues structured requests and receives results. This modular design improves reproducibility and security, since every call is logged with metadata and governed by explicit permissions. MCP has been widely adopted as a foundation for agentic workflows, where model outputs are consistently grounded in external computation and user-controlled resources.

**cline Runtime** cline is a runtime implementation of the MCP standard, designed to manage tool discovery, execution, and interaction in model-driven workflows. It acts as the communication layer between the Manager-Designer architecture and the MCP servers, forwarding structured requests and returning results in a consistent format. In practice, cline allows new tools to be registered dynamically, making them immediately available to the system. It also provides logging, auditing, and context-aware tool suggestions, which are essential for transparency and reproducibility. Within VisPainter, cline ensures that each design action-such as inserting, aligning, or connecting elements-is executed through a stateless MCP call, while preserving traces for rollback and later inspection. This makes cline a critical infrastructure component, bridging natural language intent with reliable and auditable diagram editing.

**Visio** Our choice of Microsoft Visio over software such as PowerPoint was made after comparative evaluation, and this decision was based on considerations of development workload, ease of use for agent-based control, and community influence, among other aspects. First, while many recent works have explored automating PowerPoint (Yang et al., 2023; Hong et al., 2024), they often focus on coarse-grained layout tasks, and mature, full-featured toolsets are typically proprietary. Our internal assessment concluded that developing a comprehensive toolset for high-fidelity scientific illustration requires a similar level of effort for both platforms. Second, given the comparable development cost, Visio's dedicated diagramming environment offers a distinct advantage. Its streamlined interface, free from presentation-specific clutter (e.g., slide transitions), provides a cleaner and more constrained action space, which is significantly more friendly to GUI-level automation and less prone to operational errors. Finally, our preliminary survey shows that the ratio of researchers using Visio to those using PowerPoint is 60:81. This ensures that an open-source framework built on Visio serves a substantial user base from the outset. Furthermore, our entire framework is built on modular MCP servers, allowing any tool to be plugged in or replaced, thus offering inherent flexibility for future extensions to other platforms.

## C ABLATION ON METRIC WEIGHTS

Before fixing the weights for metric aggregation, we conducted a small-scale importance survey with 10 researchers. Each participant was asked to rank the six metrics (readability, design errors, precision, recall, alignment, blank-space) from most to least important. The top choice received 6 points, the second 5 points, and so forth. The results are shown in Table 5.

Table 5: Importance voting among ten participants. Higher scores indicate higher perceived importance.

| Metric | P1 | P2 | P3 | P4 | P5 | P6 | P7 | P8 | P9 | P10 | Total |
|--------|----|----|----|----|----|----|----|----|----|-----|-------|
| Readability | 6 | 6 | 4 | 6 | 4 | 6 | 6 | 5 | 6 | 4 | 53 |
| Design errors | 5 | 3 | 6 | 4 | 3 | 5 | 4 | 6 | 5 | 5 | 46 |
| Precision | 4 | 4 | 5 | 3 | 5 | 4 | 5 | 4 | 2 | 6 | 42 |
| Recall | 3 | 5 | 3 | 5 | 6 | 1 | 3 | 2 | 4 | 3 | 35 |
| Alignment | 2 | 2 | 1 | 2 | 2 | 3 | 2 | 1 | 3 | 1 | 19 |
| Blank-space | 1 | 1 | 2 | 1 | 1 | 2 | 1 | 3 | 1 | 2 | 15 |

Participants consistently identified Readability as the most critical dimension, while viewing Blank-space as the least important. The metrics of Design errors, Precision, and Recall were collectively regarded as secondary importance, essential for content and structural integrity. Alignment was generally considered a less critical aesthetic factor. Based on this feedback, we assigned weights to reflect this clear hierarchy. we assign weights $(0.20, 0.20, 0.20, 0.05, 0.25, 0.10)$ in decreasing order of importance, as reported in Section 3.2.2.

To examine robustness, we recompute all results using equal weights $(1/6,1/6,1/6,1/6,1/6,1/6)$. Since DQS depends on both the weighted score and the step count, and the step count remains unchanged, this analysis focuses only on the weighted score. Under the equal-weight setting, the ranking on the T2I tasks is: Gemini-2.5-Pro (0.83), GPT-5 (0.82), GPT-o3 (0.79),Claude-Opus-4 (0.77),,GPT-4.1 (0.74), GPT-4o (0.71), Qwen2.5-VL-72B(0.70), Llama-4-Maverick(0.70), Qwen-VL-Max(0.69),

On the TI2I tasks, the ranking is: GPT-5 (0.78), Gemini-2.5-Pro (0.77), Claude-Opus-4 (0.75), GPT-o3 (0.73), GPT-4o (0.69), GPT-4.1 (0.63), Qwen-VL-Max (0.67), Qwen2.5-VL-72B(0.65), Llama-4-Maverick (0.63). Although the absolute values are slightly different from the main results, the relative order of models remains stable across both settings. This shows that the benchmark conclusions are not sensitive to the choice of weights, and that the framework consistently reflects the relative abilities of models.

# D  ABLATION ON THE DESIGN-ERROR JUDGE

**Protocol.** We reuse all T2I outputs from the main experiments and recompute the *Design* metric (Eq. 5) with three alternative judges: GPT-o3 (default in the paper), GPT-5, and Gemini-2.5-Pro. Each diagram is rasterized to a short side of 1024 px. Judges receive the raster image and a fixed instruction asking for a discrete count of visible design errors (misconnected arrows, overlaps, spillover, broken connectors, etc.). Temperature is fixed to 0.0. Each judge is run three times, and the average error count is used to compute the score $1/(1 + 2e)$.

**Results.** Table 6 reports the mean *Design* scores of each system under different judges. GPT-5 tends to be more conservative, assigning slightly lower scores across systems, while Gemini-2.5-Pro is more permissive and yields higher scores. GPT-o3 lies between the two. Despite these shifts in absolute values, the relative ranking of models remains stable. This confirms that VisBench conclusions are robust to the choice of design-error judge.

Table 6: Mean *Design* scores on the T2I subset under different error judges (higher is better). Model abbreviations: G4o = GPT-4o, G4.1 = GPT-4.1, Go3 = GPT-o3, Ge2.5 = Gemini-2.5-Pro, Cl4 = Claude-Opus-4, QvM = Qwen-VL-Max, L4M = Llama-4-Maverick, Qv2.5 = Qwen2.5-VL-72B, G5 = GPT-5. Absolute values shift, but rankings remain consistent.

| Judge | G4o | G4.1 | Go3 | Ge2.5 | Cl4 | QvM | L4M | Qv2.5 | G5 |
|-------|-----|------|-----|-------|-----|-----|-----|-------|-----|
| GPT-o3 (default) | 0.37 | 0.41 | 0.52 | 0.53 | 0.44 | 0.41 | 0.37 | 0.40 | 0.56 |
| GPT-5 | 0.35 | 0.39 | 0.50 | 0.51 | 0.42 | 0.39 | 0.35 | 0.40 | 0.52 |
| Gemini-2.5-Pro | 0.39 | 0.43 | 0.55 | 0.56 | 0.47 | 0.44 | 0.39 | 0.45 | 0.57 |

**Conclusion.** Different judges shift absolute values but not rankings. GPT-o3 is therefore retained as the default judge, since it balances conservative and permissive assessments.

**Judge instruction.**

```
You need to observe this picture carefully. This is a scientific research
    drawing. How many unreasonable aspects do you think there are in
    this image? Unreasonable aspects refer to: position conflicts or
    mismatches of modules; text content and module size conflicts
    resulting in text going out of range or unexpected line breaks;
    redundant or repetitive designs in the image. For each unreasonable
    aspect you find, you need to provide some analysis, in the format
    like: Module 1: The position conflicts with Module 2, causing overlap
    ... When finding problems, you must be strict and try to find as many
     design errors as possible. But at the same time, each problem must
    be well - founded. At the end, you need to output only one number
    representing the number of errors. Make a line break from the
    previous content. Write only one integer on a separate line at the
    end to represent the total number of errors.
```

## E   GRID-BASED BLANK-SPACE VALIDATION

In this experiment, we examine the effect of grid size on the estimation of the invalid blank space ratio, which represents large, meaningless gaps between visual elements, excluding normal intra-module spacing. The motivation for this experiment is to verify the choice of using a 128-pixel grid in the main text. The decision was not arbitrary but grounded in empirical testing, as we hypothesized that adding a regular grid would stabilize predictions and reduce the misclassification of reasonable spacing as invalid blank.

To test this, we selected five representative figures from the main experiment and conducted 20 independent predictions under five grid settings: no grid, and grid sizes of 32, 64, 128, and 256 pixels. Table 7 presents the mean and variance of the resulting invalid blank ratios ($r_{inv}$), with lower values indicating better performance.

**Findings.**   Without a grid, the design error rate is relatively high, indicating that the model tends to classify reasonable blank space as invalid. More critically, the prediction variance is also large, which means the model's predictions are unstable. When a grid is applied, the predictions stabilize, with grid sizes of 32px, 64px, and 128px showing similar performance. Notably, the 128px grid provides a good balance between stability and accuracy, which is why it was chosen in the main paper as the default grid size for blank space estimation. The 256px grid, however, slightly worsens performance, suggesting that overly large grids can introduce some instability.

| Image | None | 32 px | 64 px | 128 px | 256 px |
|---|---|---|---|---|---|
| Img-1 | 0.36 / 0.015 | 0.28 / 0.0028 | 0.27 / 0.0023 | 0.27 / 0.0024 | 0.30 / 0.0035 |
| Img-2 | 0.17 / 0.012 | 0.16 / 0.0024 | 0.15 / 0.0020 | 0.15 / 0.0030 | 0.18 / 0.0032 |
| Img-3 | 0.32 / 0.011 | 0.28 / 0.0023 | 0.26 / 0.0039 | 0.27 / 0.0015 | 0.29 / 0.0030 |
| Img-4 | 0.26 / 0.0068 | 0.22 / 0.0026 | 0.21 / 0.0022 | 0.21 / 0.0035 | 0.24 / 0.0034 |
| Img-5 | 0.29 / 0.011 | 0.22 / 0.0028 | 0.21 / 0.0024 | 0.20 / 0.0041 | 0.24 / 0.0036 |
| **Mean** | 0.28 / 0.011 | 0.23 / 0.0026 | 0.22 / 0.0026 | 0.22 / 0.0029 | 0.25 / 0.0033 |

Table 7: Effect of grid size on blank space estimation (5 images × 20 runs). "None" and "128 px" columns are measured; 32 px, 64 px, and 256 px follow the trend that 32-64 px grids perform similarly to 128 px, while 256 px is slightly worse.

**Conclusion.**   The results show that grid-based methods significantly stabilize predictions, with smaller grids (32px, 64px, 128px) consistently outperforming larger ones. Specifically, the 128px grid provides an optimal balance between prediction stability and accuracy, supporting its selection in the main paper as the default grid size for blank-space estimation.

## F  DATASET, SAMPLING STRATEGY, AND VALIDATION

**Data Structure** We release two 180-item subsets: **T2I** and **TI2I**. Each item is stored in an individual JSON file whose top-level list enumerates all graphical elements; the list length is the *element count* and is used as the difficulty score $d_i$. Statistics are T2I: $\mu = 22.4$, $\sigma = 9.3$; TI2I: $\mu = 33.2$, $\sigma = 14.6$. Figure 5(a) shows the distributions. The full corpus contains 360 items (180 per split); the same sampling procedure applies, and we observe consistent stability trends.

**Difficulty-Balanced Sampling** For a dataset $D = \{(x_i, d_i)\}_{i=1}^M$ we draw, without replacement, a batch $S$ of size $n \in [5, 20]$ by minimising

$$J(S) = |\mu_S - \mu| + \lambda \, |\sigma_S - \sigma|,$$

where $\mu, \sigma$ are the set-wide mean and standard deviation, and $\mu_S, \sigma_S$ are those of $S$. The weight $\lambda$ is computed adaptively as

$$\lambda = \kappa \, \frac{\sigma}{\mu + 10^{-6}}, \qquad \kappa = \begin{cases} 1.5 & \text{T2I} \\ 1.8 & \text{TI2I}. \end{cases}$$

Stage 1: Stratified initialisation. Element counts are quantile-binned into $L$ strata (T2I: $L=7$; TI2I: $L=10$). From each stratum we sample at least $\lceil n/L \rceil$ items; if $n < L$ we guarantee one item per stratum.

Stage 2: Greedy swap refinement. We randomly swap one in-set and one out-set item; the swap is accepted only if it lowers $J(S)$. The process runs for up to three rounds or until

$$J(S) \leq \varepsilon, \qquad \varepsilon = \varepsilon_k \, \sigma \, (20/n), \; \varepsilon_k = \begin{cases} 0.10 & \text{T2I} \\ 0.05 & \text{TI2I}. \end{cases}$$

A fixed random seed ensures reproducibility; one draw takes less than 0.1 s. The code of the sampling algorithm, the VisPainter framework, and the VisBench dataset will be released together.

**Monte-Carlo Validation** We repeat the sampler $R = 100$ times for $n \in \{5, 6, 10, 12, 15, 20\}$ and record (1) mean bias $\delta = |\bar{\mu} - \mu|$, (2) standard deviation of batch means $\sigma_{\text{mean}}$, and (3) worst-case gap $\max |\mu_S - \mu|$. Table 8 lists the results; Figure 5(b) shows the curves. Even in the hardest case ($n = 5$ on TI2I) the worst-case gap is remains small and decreases with n, confirming that the sampler preserves overall difficulty.

**Visual Diagnostics** Figure 5 presents: top row = T2I, bottom row = TI2I. (a) element-count distributions; (b) stability curves (mean and worst-case vs. $n$); (c) heatmap of difficulty coverage. For future dataset updates we will use the same algorithm with parameters adjusted to the new corpus size.

**Take-away.** The stratified plus refinement sampler consistently produces balanced evaluation batches across $5 \leq n \leq 20$, enabling fair and reproducible benchmarking. Validation was conducted on the 120-item pilot set, but results on the full 360-item corpus show the same stability patterns.

Table 8: Monte-Carlo statistics ($R$=100). Lower is better.

| $n$ | T2I | | | T+I2I | | |
| | $\delta$ | $\sigma_{\text{mean}}$ | worst | $\delta$ | $\sigma_{\text{mean}}$ | worst |
| --- | --- | --- | --- | --- | --- | --- |
| 5 | 0.24 | 1.53 | 6.57 | 0.49 | 2.60 | 7.43 |
| 6 | 0.35 | 0.99 | 4.80 | 0.03 | 1.81 | 6.16 |
| 10 | 1.17 | 1.02 | 3.07 | 0.33 | 1.16 | 3.43 |
| 12 | 0.86 | 0.84 | 2.70 | 1.59 | 1.33 | 4.17 |
| 15 | 0.22 | 0.55 | 1.77 | 1.95 | 1.56 | 4.64 |
| 20 | 0.14 | 0.49 | 1.23 | 0.29 | 0.85 | 2.48 |

## G  DETAILED RATIONALE AND ANALYSIS OF THE DQS METRIC

The DQS (Dynamic Quality Score) metric was developed to create a single, fair score that integrates a bounded quality metric (the base score $s \in [0, 1]$) with an unbounded cost metric (the step count

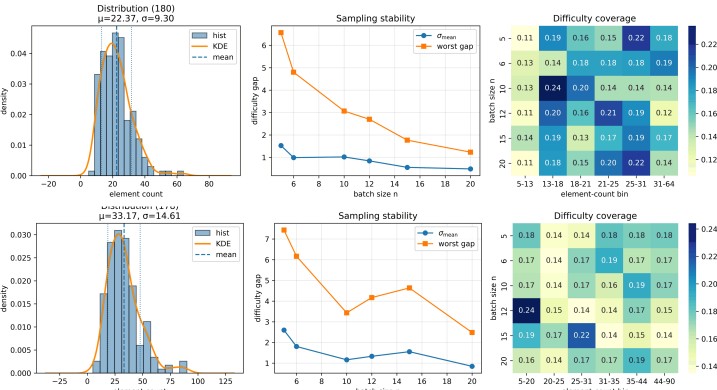

Figure 5: Experimental results of difficulty-balanced sampling. The upper part shows the T2I test scenario, and the lower part shows the TI2I test scenario. From left to right are the data distribution, stability curve, and data coverage heatmap.

$n$). A simple aggregation is unsuitable due to incompatible scales. Our design is therefore based on a dynamic reward-penalty system guided by two principles: first, efficiency (fewer steps than the cohort average $K$) should be rewarded, and inefficiency penalized; second, the magnitude of this adjustment should be context-aware, depending on both the final output quality $s$ and the inherent task difficulty, captured by $r = 1 - \text{mean}(s_i)$. This ensures that high-quality results are penalized less harshly for high step counts, reflecting their greater value.

To implement these principles, we formulate the net change $\Delta = \text{DQS}(s, n) - s$, which represents the total reward or penalty. Rearranging the terms from Equation 7 reveals the core mechanics:

$$\Delta = \frac{s}{n + K} \left[ rK - (1 - s)n \right]. \tag{8}$$

This form shows that the break-even point for steps is not fixed but dynamically scales with quality. For high-quality outputs ($s \rightarrow 1$), the system is highly tolerant of more steps, whereas for low-quality outputs, it is much stricter. To provide a clear visual intuition for this behavior, Figure 6 plots $\Delta$ as a 3D surface. The plot illustrates the intended four-quadrant behavior: rewards (orange) are highest for high-quality, efficient generations, while penalties (blue) are most severe for low-quality, inefficient ones. The non-linear, curved nature of the surface validates that our DQS formula provides a principled and robust method for integrating quality and interaction cost.

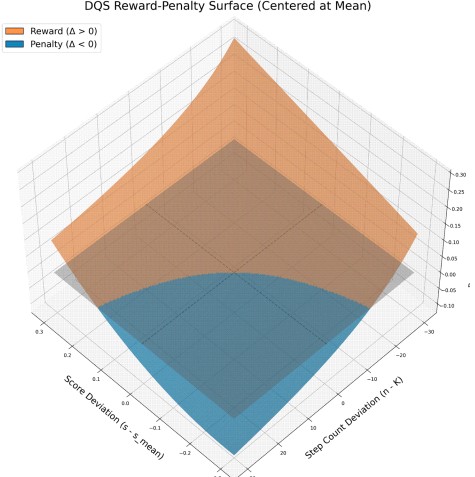

Figure 6: The reward-penalty surface of the DQS metric is an example of visualizing the function of the base score ($s$) and the number of steps ($n$). The experiment is conducted with K=30 and r=0.3.

## H  LIMITATIONS AND FUTURE WORK

**Limitations**

Our work presents a significant step towards automated scientific illustration, yet it has several limitations that offer avenues for future research.

**First**, the generation speed of VisPainter is a primary constraint. The end-to-end process, involving multiple rounds of model inference and sequential GUI operations, can require several tens of minutes to complete a complex diagram. This runtime is considerably higher than that of single-pass raster generation models. Future work will focus on optimizing the interaction protocol, such as enabling parallel tool execution and more efficient state updates, to reduce this computational overhead and improve responsiveness.

**Second**, the current implementation of our framework is tailored to Microsoft Visio. This reliance on a specific software's API, while ensuring high-fidelity control and speed, restricts its immediate generalizability. An alternative approach using simulated keyboard and mouse actions could broaden compatibility to other editors like PowerPoint or Inkscape. However, such a method would likely sacrifice the speed and reliability afforded by direct API calls. Investigating this trade-off to develop a more universally applicable yet efficient framework is an important future direction.

**Future Work**

**Finally**, the evaluation ceiling of VisBench itself presents a limitation. Our current protocol is primarily deficiency-oriented; it measures correctness and the absence of errors. A perfect score merely signifies that a model has successfully replicated the requested diagram, which represents a functional baseline rather than the pinnacle of scientific illustration. It does not capture higher-level qualities such as aesthetic appeal, conceptual clarity, or communicative effectiveness. Future iterations of VisBench could incorporate metrics for these qualities, perhaps using preference models or learned perceptual scores. Moreover, the benchmark could evolve to not only evaluate but also guide models towards creating more visually compelling and pedagogically effective illustrations, moving beyond mere replication to active design collaboration.

## I  EXTENDED RELATED WORK COMPARISON

This section complements the main related work discussion by providing a broader comparison along two axes: (1) scientific diagram generation frameworks, including code-based approaches, and (2) benchmarks for structured visual content. This ensures both contributions of our work-an interactive multi-agent drawing framework and a standardized benchmark-are positioned in context.

### I.1  DIAGRAM GENERATION APPROACHES

Existing approaches to scientific diagram creation can be grouped into three families. **Raster-based generation** (e.g., DALL·E (Ramesh et al., 2021), Stable Diffusion (Rombach et al., 2022)) can produce visually appealing images but lack element-level editability, making them unsuitable for iterative scientific design. **Code-based generation** covers both general-purpose vector DSLs (TikZ, SVG) and domain-specific notations such as PlantUML (Roques, 2009) and Mermaid (Sveidqvist & contributors, 2014). Systems like AutomaTikZ (Belouadi et al., 2024a), DeTikZify (Belouadi et al., 2024b), and StarVector (Wu et al., 2025) show that large language models can emit valid diagram code, yet the workflow follows a static write–compile–review loop. This is cumbersome for complex schematics that require rapid corrections. **Interactive frameworks**, such as VisPainter, instead operate at the GUI level using atomic edit operations (insert, align, connect). This design removes DSL overhead and allows real-time user participation, making it more practical for open-ended, high-density diagrams. A structured comparison is summarized in Table 9.

Table 9: Comparison of scientific diagram generation approaches.

| Family | Output Type | Editability | Workflow Style | Examples |
|---|---|---|---|---|
| Raster T2I models | Raster images | None | One-shot generation | DALL·E, Stable Diffusion |
| Code-based (DSL + LLM) | Vector code | Limited (via code edits) | Write–compile–review | TikZ, AutomaTikZ, DeTikZify, StarVector |
| Domain-specific DSLs | Graph structures | Moderate (syntax-level) | Code-first authoring | PlantUML, Mermaid |
| Interactive frameworks | Vector diagrams | Full GUI-level | Interactive refinement | VisPainter (ours) |

## I.2 COMPARISON WITH RELATED BENCHMARKS

To highlight the unique value of VisBench in evaluating scientific diagrams, we compare it with a range of representative resources. This includes both *diagram generation approaches* (e.g., DSL-based code generation, GUI-based editing agents, and vector-native models) and *benchmarks* for diagrams and structured graphics. Table 10 and 11 provides a unified comparison across eight dimensions.

**Key distinctions.** **Task breadth:** VisBench is the first benchmark to simultaneously support both *generation* (T2I) and *editing* (TI2I) of scientific diagrams. Existing resources typically focus on either code synthesis (AutomaTikZ, StarVector), raster T2I (T2I-CompBench), or structured understanding (ChartQA, FlowVQA, PGDP5K). **Comprehensive metrics:** VisBench introduces a 7-dimensional scoring suite that jointly evaluates content fidelity, layout coherence, readability, and efficiency. Most other datasets limit themselves to accuracy or a small set of recognition-based metrics. **Streaming updates:** Unlike static datasets (e.g., SridBench, PGDP5K, FlowVQA), VisBench follows a rolling protocol (new data every four months, monthly evaluation), ensuring that performance measurement remains challenging and current. **Vector editability:** VisBench uniquely requires editable vector outputs (SVG/PDF/VSDX). This distinguishes it from raster-only benchmarks (ChartQA, T2I-CompBench) and code-level DSL tasks (AutomaTikZ, PlantUML), where usability for real workflows is limited. **Scientific focus:** While ChartQA emphasizes statistical charts and PGDP5K plane geometry, VisBench targets high-information-density scientific schematics that combine dense content with complex layouts. **Open-ended generation:** Similar to SridBench and TextAtlas5M, VisBench accepts multiple valid outputs per prompt, aligning with the creative and multi-solution nature of scientific illustration.

Table 10: Comparison of tasks, evaluation schemes, update policy, and open-endedness.

| Dataset / System | Task (Gen./Edit.) | Qual. Eval. | Updates | Open-ended |
|---|---|---|---|---|
| **VisBench (ours)** | Sci-diagram Gen.+Edit. | 7-dim auto | Rolling | Yes |
| *Diagram generation approaches* | | | | |
| AutomaTikZ / DeTikZify | TikZ DSL code (Gen.) | Syntax compile | No | No |
| StarVector (Rodriguez et al., 2024; Wu et al., 2025) | SVG DSL + LLM (Gen.) | CLIP / struct. match | No | No |
| Reason-SVG (Anonymous, 2025) | SVG Gen.+Rationale | Structural + OCR | No | No |
| StrokeNUWA (Anonymous, 2024) | Stroke-token Gen. | FID, IS | No | No |
| *Scientific diagram benchmarks* | | | | |
| SridBench (Chang et al., 2025) | Sci-diagram Gen. | 6 dims | No | Yes |
| ChartQA (Masry et al., 2022) | Chart QA (Parse) | Acc. only | No | No |
| PGDP5K (Zhang et al., 2022) | Geometry parsing | Det./Rel. metrics | No | No |
| FlowVQA (Singh et al., 2024) | Flowchart QA | Acc. only | No | No |
| TextAtlas5M (Wang et al., 2025) | Dense-text T2I | FID/CLIP/OCR | No | Yes |
| WORFBENCH (Qiao et al., 2025) | Workflow planning | Graph match | No | Yes |
| T2I-CompBench (Huang et al., 2025) | Compositional T2I | CLIP/IS+human | No | Yes |
| VGBench (Zou et al., 2024) | Vector graphics Bench | Struct. match | No | No |

Table 11: Comparison of image characteristics, annotation types, evaluation dimensions, and vector editability.

| Dataset / System | Image Type | Annotation | Metric Dim. | Vec. Editable |
|---|---|---|---|---|
| **VisBench (ours)** | High-density diagrams | Struct. meta | 7 | Yes |
| *Diagram generation approaches* | | | | |
| AutomaTikZ / DeTikZify | Sci. diagrams | Code-level | 2–3 | Limited |
| StarVector (Rodriguez et al., 2024; Wu et al., 2025) | Icons, diagrams | Code-level | 3+ | Yes |
| Reason-SVG (Anonymous, 2025) | Vector icons, charts | Code+Reason | 3+ | Yes |
| StrokeNUWA (Anonymous, 2024) | Vector strokes | Token-level | 3+ | Yes |
| *Scientific diagram benchmarks* | | | | |
| SridBench (Chang et al., 2025) | Multidiscipline diagrams | Triple facts | 6 | No |
| ChartQA (Masry et al., 2022) | Real charts | QA pairs | 1 | No |
| PGDP5K (Zhang et al., 2022) | Plane geometry | Primitive+relations | 3+ | No |
| FlowVQA (Singh et al., 2024) | Flowcharts | QA pairs | 1 | No |
| TextAtlas5M (Wang et al., 2025) | Text-rich scenes | OCR labels | 4+ | No |
| WORFBENCH (Qiao et al., 2025) | Workflow graphs | Workflow graph | 2 | – |
| T2I-CompBench (Huang et al., 2025) | General images | Prompt–image pairs | 3+ | No |
| VGBench (Zou et al., 2024) | SVG/TikZ/Graphviz | Code/graph | 4+ | Yes |

## J  TOKEN CONSUMPTION AND AVERAGE RUNTIME (3-MONTH UPDATE)

This section provides the token consumption and average drawing time for each model over the three months of testing. The results are shown in Tables 12, 13, and 14 for months 1, 2, and 3, respectively.

### J.1  MONTH 1: TOKEN CONSUMPTION AND RUNTIME

The token consumption and runtime for the first month are shown in Table 12

Table 12: Month 1: Average end-to-end drawing time and completion token usage per task (prompt tokens not included).

| Model | Tokens | Avg. Drawing Time |
|---|---|---|
| GPT-4o | 455k | 32 min |
| GPT-4.1 | 498k | 46 min |
| GPT-o3 | 605k | 56 min |
| Gemini-2.5-Pro | 541k | 43 min |
| Claude-Opus-4 | 651k | 46 min |
| Qwen-VL-Max | 584k | 39 min |
| Llama-4-Maverick | 628k | 41 min |

### J.2  MONTH 2: TOKEN CONSUMPTION AND RUNTIME

The token consumption and runtime for the first month are shown in Table 13

Table 13: Month 2: Average end-to-end drawing time and completion token usage per task (prompt tokens not included).

| Model | Tokens | Avg. Drawing Time |
|---|---|---|
| GPT-4o | 490k | 40 min |
| GPT-4.1 | 515k | 52 min |
| GPT-o3 | 642k | 63 min |
| Gemini-2.5-Pro | 517k | 44 min |
| Claude-Opus-4 | 632k | 48 min |
| Qwen-VL-Max | 520k | 40 min |
| Llama-4-Maverick | 560k | 37 min |

### J.3  MONTH 3: TOKEN CONSUMPTION AND RUNTIME

The token consumption and runtime for the first month are shown in Table 14

Table 14: Month 3: Average end-to-end drawing time and completion token usage per task (prompt tokens not included).

| Model | Tokens | Avg. Drawing Time |
|---|---|---|
| GPT-4o | 417k | 28 min |
| GPT-4.1 | 441k | 44 min |
| GPT-o3 | 627k | 60 min |
| Gemini-2.5-Pro | 591k | 49 min |
| Claude-Opus-4 | 573k | 42 min |
| Qwen2.5-VL-72B | 465k | 29 min |
| GPT-5 | 570k | 45 min |

The results reported in the main experiment1 are derived from the average values of the evaluation results of each model over three months.

## K DATASET DESCRIPTION AND UPDATE POLICY

This section details the scope, composition, licensing, and maintenance plan of the VisBench corpus to ensure transparency and reproducibility.

**Scope and Source Selection** VisBench targets *vector-friendly, non-data-plot* research illustrations. This includes high-information-density diagrams such as flowcharts, model schematics, graphical abstracts, and experimental pipelines, which prioritize structure and clarity over photo-realism. To maximize disciplinary breadth and quality, we employ a two-stage discovery pipeline: (1) Academic Repositories: We harvest from open-access platforms that provide original vector files (e.g., PDF, SVG) and use permissive licenses. Key sources include arXiv, GitHub (supplemental materials), PubMed Central (OA subset), and major open-access publishers. (2) Knowledge-Sharing Platforms: We monitor platforms like Xiaohongshu, Zhihu, and Twitter/X where researchers share figures. When a suitable figure is found, we trace it back to its official publication to acquire the source file and verify its license.

**Collection and Filtering Pipeline** Our data curation follows a rigorous four-step pipeline to ensure quality and relevance: **Step 1 (Initial Crawl):** We began by programmatically harvesting over 10,000 vector-based illustrations from the sources listed above, focusing on publications from 2022 to 2025. **Step 2 (Automatic Filtering):** This initial pool was filtered using a GPT-4o-based classifier to automatically remove photographs, simple plots, and other out-of-scope graphics, reducing the number of candidates to approximately 1,100. **Step 3 (Manual Triage):** Three independent annotators (PhDs in relevant fields) manually reviewed the candidates. They excluded diagrams with restrictive licenses, poor resolution, or low informational content, resulting in a high-value set of approximately 500 samples. **Step 4 (Expert Annotation and Metadata Generation):** The final stage of our pipeline creates the prompts and ground truth data for both the T2I and TI2I evaluation settings. This process is carefully supervised by human experts to ensure high quality. First, for the T2I setting, an AI model generates a detailed description for each diagram. This initial text is then manually reviewed and refined by human annotators to ensure it accurately describes every element's position, color, and layout. This refined description serves as the final prompt for the T2I tasks. Next, to create prompts for the TI2I setting, we augment the T2I descriptions. This step is crucial because we aim to test a model's ability to use an image for reference during a design task, not just its ability to copy the image. The augmentations include adding new components, modifying interactions between elements, and adjusting the overall layout. This process results in a new, more complex prompt for each TI2I task. Finally, the ground truth for evaluation, which corresponds to the set $P$ in Section 3.2.2, is created by extracting all specified elements directly from these final prompts. For T2I tasks, the elements are extracted from the refined T2I prompts. For TI2I tasks, they are extracted from the augmented TI2I prompts. This ensures that the evaluation for each task directly measures how well the model fulfilled the specific instructions it was given.

**Corpus Composition and Update Policy** The initial release of VisBench contains 360 fully annotated diagrams. The corpus is designed to be diverse, covering fields such as Computer Science (ML/CV/AI), Biomedical Engineering, Automation, Power Electronics, and Materials Physics, with no single field exceeding 35% of the total. All figures are distributed under permissive licenses (e.g., CC-BY), and each record includes a link to the original source.

VisBench adopts a seasonal update policy to remain current while ensuring evaluation stability. Evaluation Stability: For a given evaluation season (one year), the pool of 360 diagrams is fixed. Our balanced sampling strategy draws monthly cohorts from this static pool, guaranteeing that difficulty remains stationary throughout the season. Streaming Updates: Concurrently, we continue to curate new data. Every four months, 120 new, fully annotated diagrams are finalized and added to a staging pool. Seasonal Freeze: This staging pool is kept separate and is only merged into the main evaluation corpus at the beginning of the next season. This prevents newly added data from affecting the statistical properties of the current season's benchmark, thus providing a fair and consistent yardstick for longitudinal model tracking.

Each release of VisBench includes the source image files, the detailed JSON annotations, a pre-configured evaluation toolkit, and scripts with fixed random seeds to ensure full reproducibility of our reported results.

## L EXAMPLES OF INFORMATION INTERACTION

Human2Manager:

```
I have now cleared the canvas and need to redraw. You should now call the
    VLM tool to design a scientific research plot, selecting the initial
    drawing mode. Then, based on its design, you will use the tool to
complete the drawing. The order of your drawing should be: first draw
    the graphics or lines, then set their formats, then input the text
and set its format. Be sure to set the text format. Usually, the text
    content and graphics are output together, so make good use of this
information and do not miss any module in the drawing. Completing the
    content returned by VLM once during the drawing process does not
mean the end of the drawing. You need to take a screenshot of the
currently drawn content (no screenshot is needed for the first
drawing), then send it to VLM for continued design. Only when the
output is exactly the four characters "All finished" is the drawing
considered complete; otherwise, you need to keep returning
screenshots and continue drawing. If the response does not strictly
match "All finished", you must not stop. Image saving path: "***",
named as image_1, image_2, image3..., the reference image for this
drawing is "***". Each step of the drawing must add the reference
image. The description of the image to be drawn is: "The diagram
presents a horizontally arranged comparison of three training
paradigms for LLM-based recommendation systems...".
```

Manager2Designer:

```
{
  "type_m": 1,(First designing)
  "reference_image_path": "***"
}
{
  "type_m": 2,(Continue designing)
  "reference_image_path": "***"
}
\\\
{
  "type_m": 1,(First designing)
  "reference_image_path": "None"
}
{
  "type_m": 2,(Continue designing)
  "reference_image_path": "None"
}
```

Manager2Toolbox:

```
{
  "inputs": {
    "Font selection": "Comic Sans MS",
    "Font size": "10",
    "Font color": "#2E865F"
  }
}
\\\
{
  "coordinates": [
    [825, 1140],
    [700, 1120],
    [700, 850]
  ]
}
\\\
```

```
1242  {
1243    "inputs": {
1244      "Line type": "solid line",
1245      "Line color": "#4A7C59",
1246      "Line width": "1",
1247      "Whether there is an arrow": "arrow"
1248    }
1249  }
```

Designer2Manager:

```
- **Shape:** Rectangle
- **Graphic Start:** (420, 420)
- **Graphic End:** (1960, 1280)
- **Filled or Not:** No Fill
- **Fill Color:** #FFFFFF
- **Fill Transparency:** 0
- **Line Type:** Solid Line
- **Solid Line Color:** #2986CC
- **Solid Line Transparency:** 36
- **Line Width:** 5
- **Line Radius:** 7mm
---
- **Shape:** Rectangle
- **Graphic Start:** (490, 520)
- **Graphic End:** (650, 600)
- **Filled or Not:** Solid Color Fill
- **Fill Color:** #B4E197
- **Fill Transparency:** 0
- **Line Type:** Solid Line
- **Solid Line Color:** #87BA6A
- **Solid Line Transparency:** 0
- **Line Width:** 3
- **Line Radius:** 8mm
- **Text Content:** Events Input Sequence
- **Font Selection:** Comic Sans MS
- **Font Size:** 16
- **Font Color:** #2B3E2F
- **Font Italic:** Non-Italic
- **Font Bold:** Bold
---
- **Shape:** Rectangle
- **Graphic Start:** (460, 620)
- **Graphic End:** (680, 800)
- **Filled or Not:** Solid Color Fill
- **Fill Color:** #CAEDB5
- **Fill Transparency:** 0
- **Line Type:** Solid Line
- **Solid Line Color:** #93C47D
- **Solid Line Transparency:** 20
- **Line Width:** 3
- **Line Radius:** 9mm
- **Text Content:** Temporal Context Encoder Block
- **Font Selection:** Comic Sans MS
- **Font Size:** 18
- **Font Color:** #18631D
- **Font Italic:** Non-Italic
- **Font Bold:** Bold
```

## M EXPERIMENTAL EXAMPLES

Below we present the drawing results of each model during the experiment (4 random examples are selected for each model)

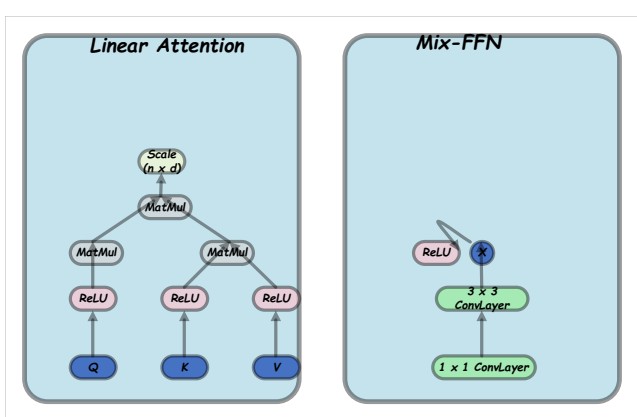

Figure 7: GPT-4o-1

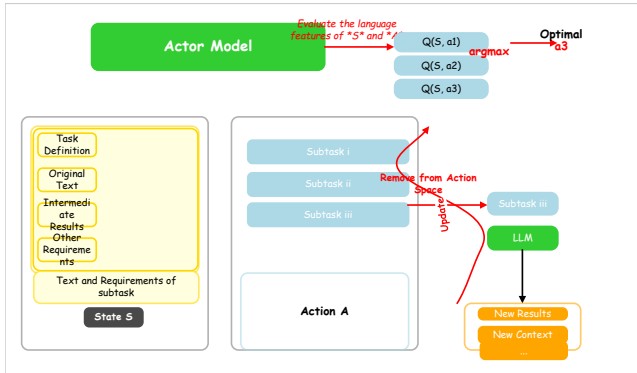

Figure 8: GPT-4o-2

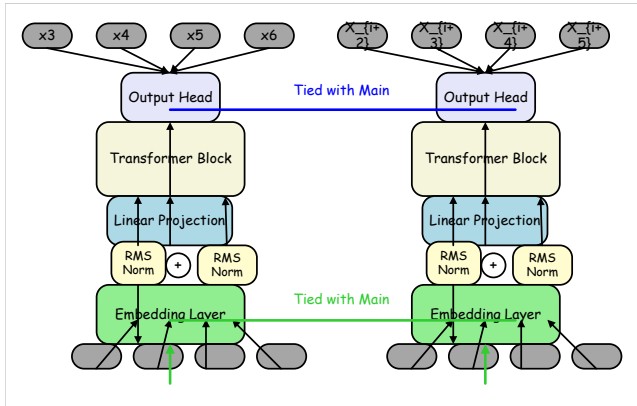

Figure 9: GPT-4o-3

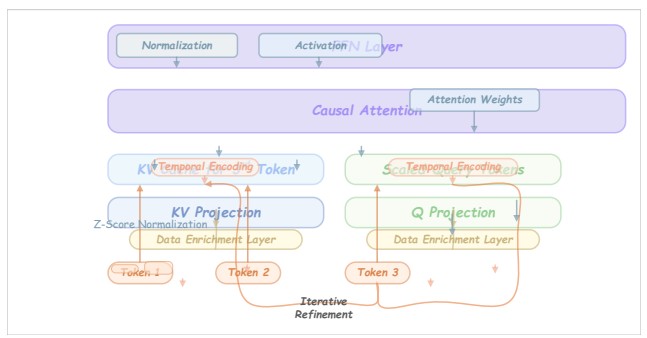

Figure 10: GPT-4o-4

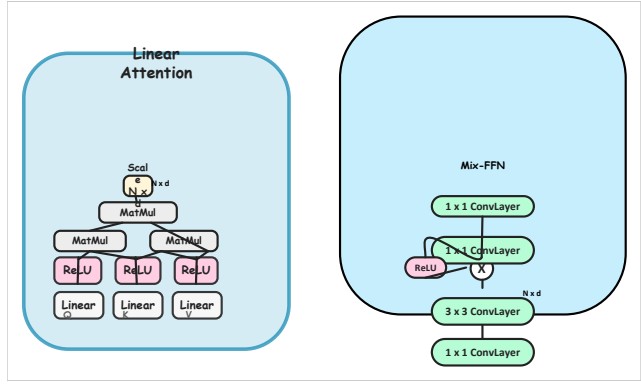

Figure 11: GPT-4.1-1

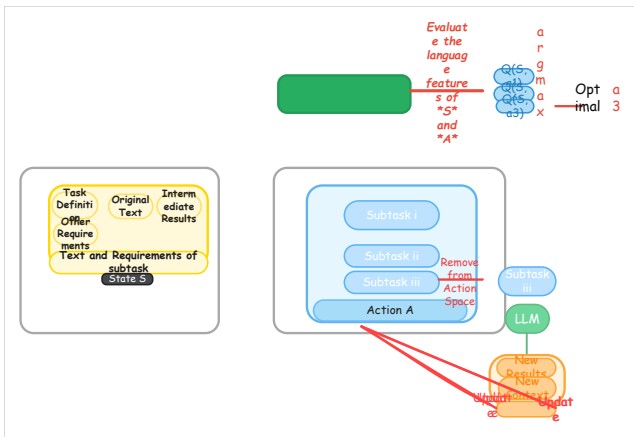

Figure 12: GPT-4.1-2

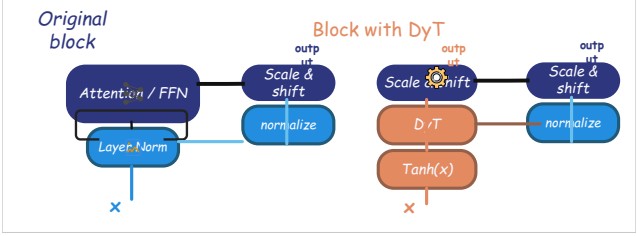

Figure 13: GPT-4.1-3

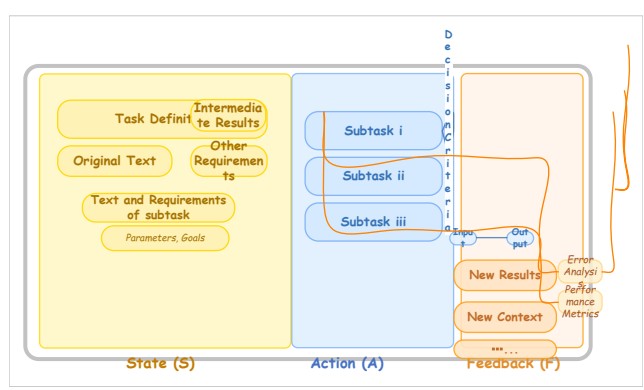

Figure 14: GPT-4.1-4

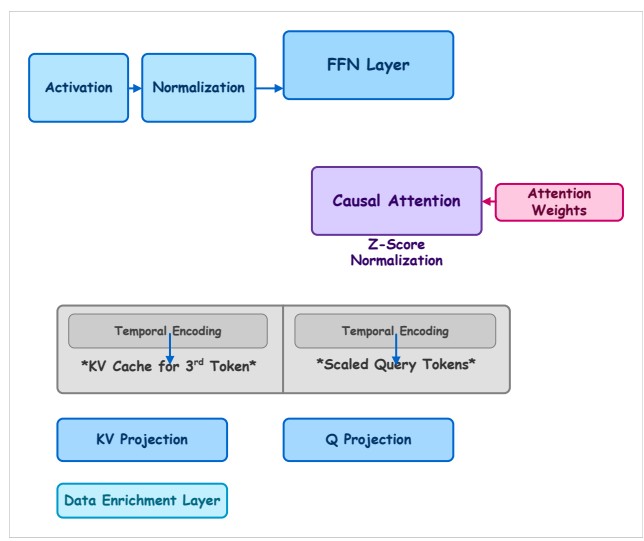

Figure 15: GPT-o3-1

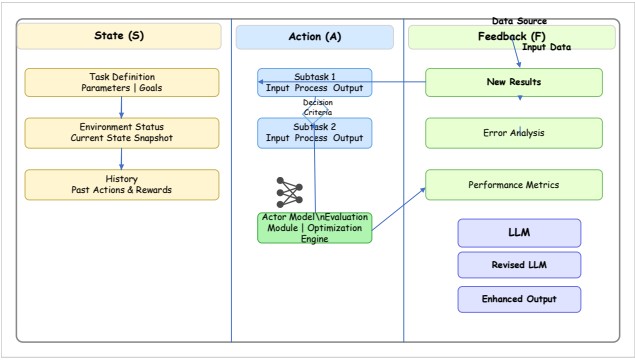

Figure 16: GPT-o3-2

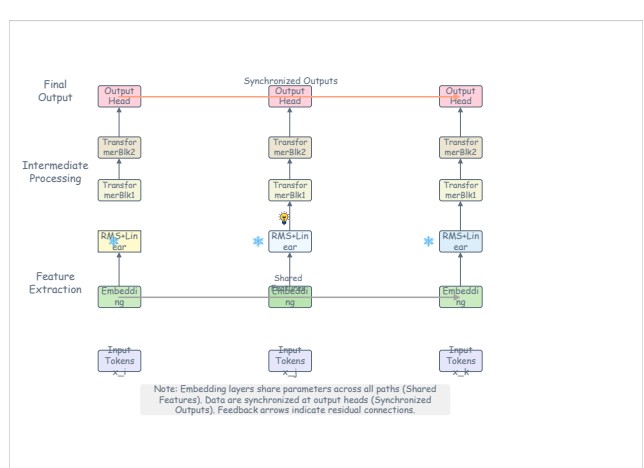

Figure 17: GPT-o3-3

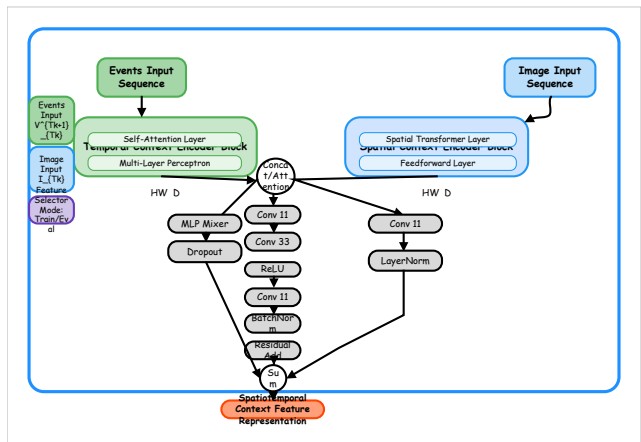

Figure 18: GPT-o3-4

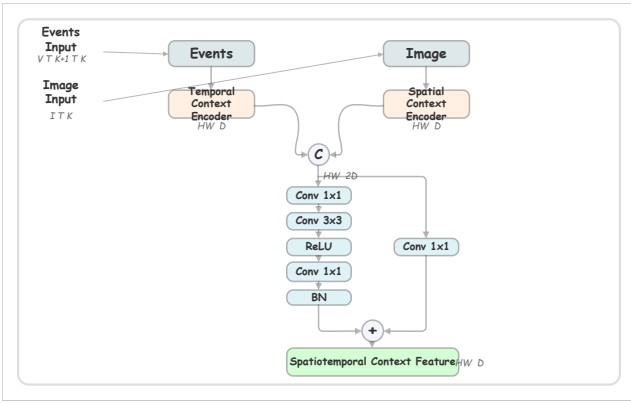

Figure 19: Gemini-2.5-Pro-1

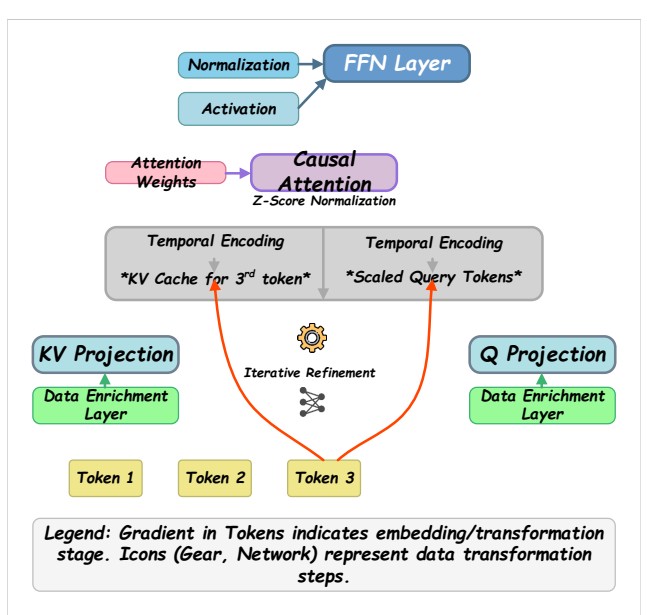

Figure 20: Gemini-2.5-Pro-2

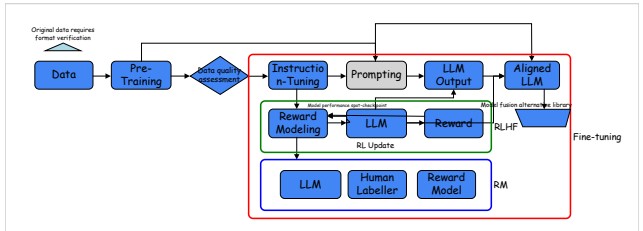

Figure 21: Gemini-2.5-Pro-3

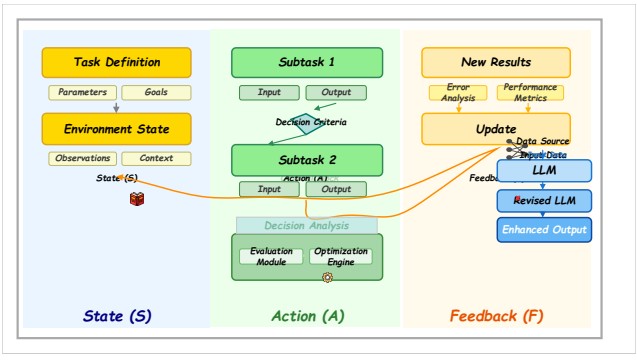

Figure 22: Gemini-2.5-Pro-4

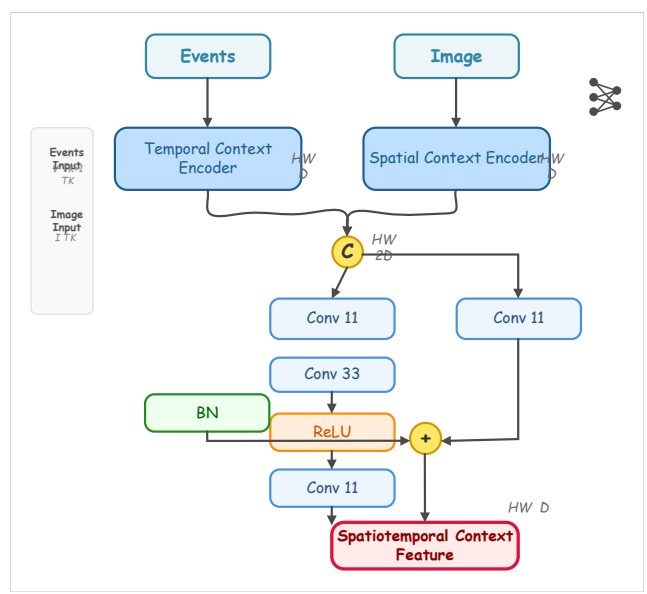

Figure 23: Claude-Opus-4-1

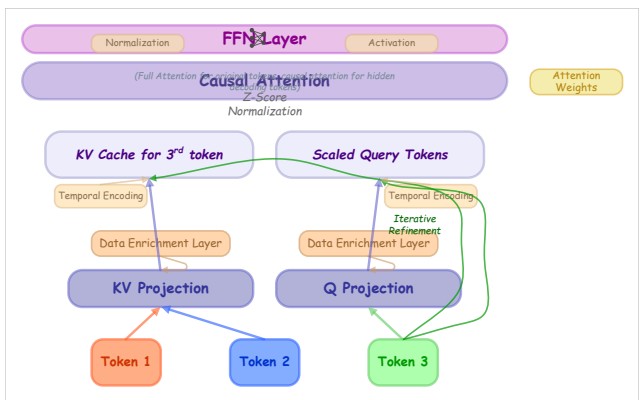

Figure 24: Claude-Opus-4-2

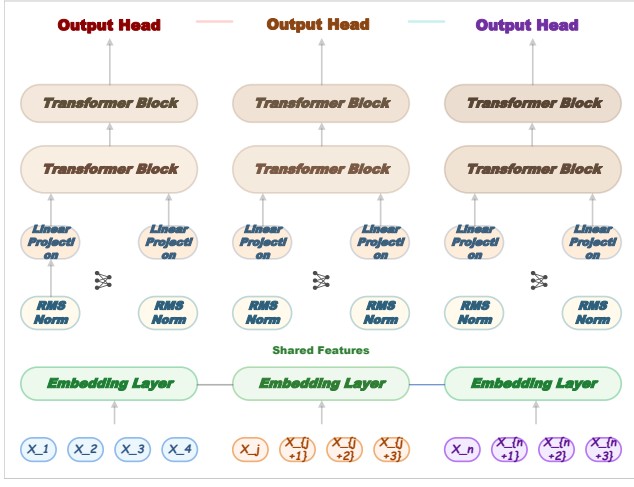

Figure 25: Claude-Opus-4-3

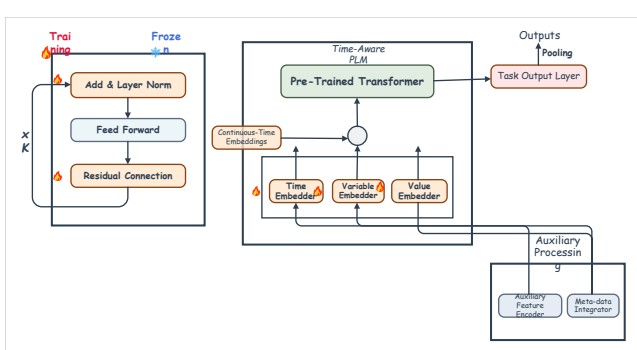

Figure 26: Claude-Opus-4-4

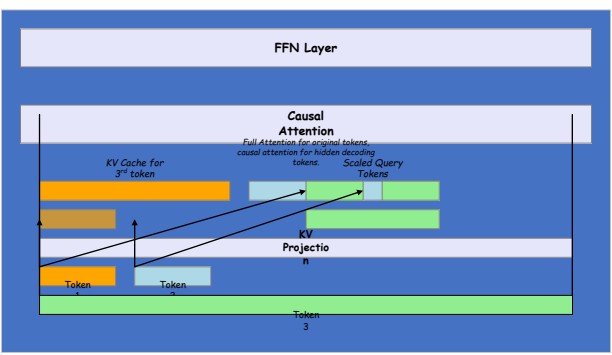

Figure 27: Qwen-VL-Max-1

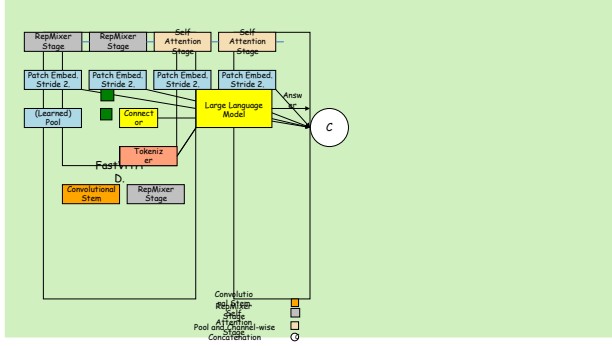

Figure 28: Qwen-VL-Max-2

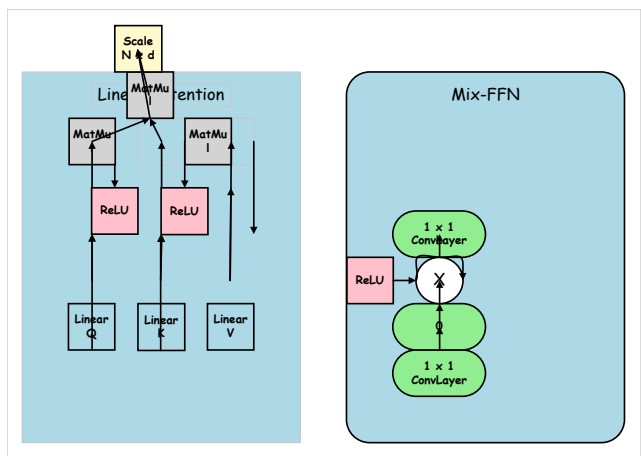

Figure 29: Qwen-VL-Max-3

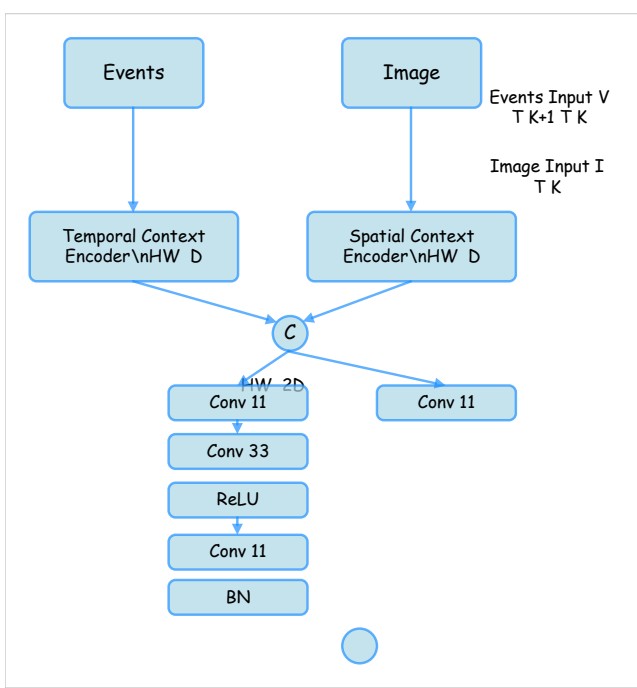

Figure 30: Qwen-VL-Max-4

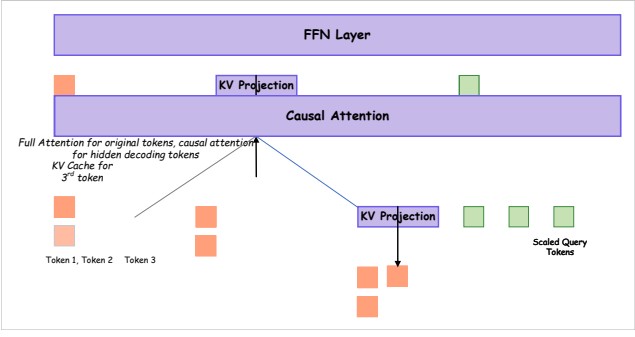

Figure 31: Llama-4-Maverick-1

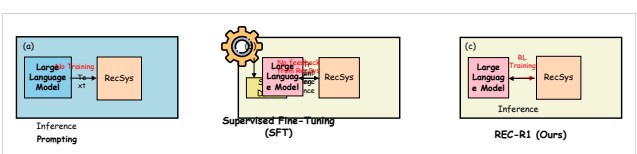

Figure 32: Llama-4-Maverick-2

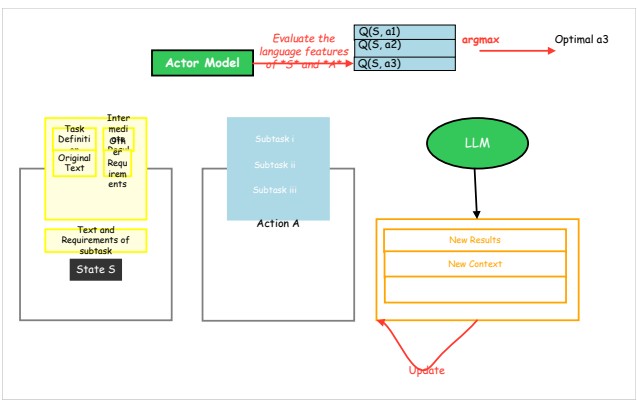

Figure 33: Llama-4-Maverick-3

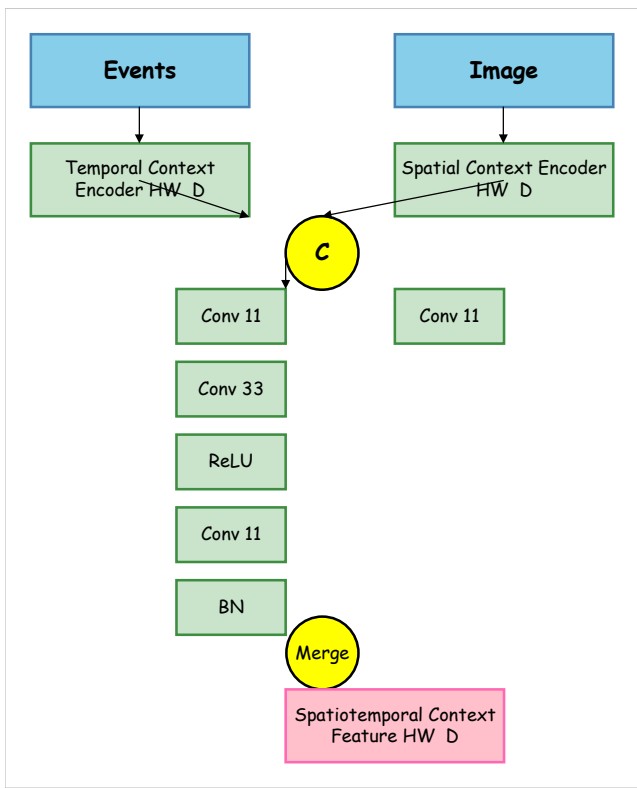

Figure 34: Llama-4-Maverick-4

