# OpenReview forum: "From Pixels to Paths: A Multi-Agent Framework for Editable Scientific Illustration"
_ICLR.cc/2026/Conference — Submitted to ICLR 2026_

### Official Review · Reviewer_GfaX · 2025-10-29

**Soundness:** 2
**Presentation:** 3
**Contribution:** 2
**Rating:** 2
**Confidence:** 3

**Summary:**

This paper presents VisPainter, a training-free multi-agent framework that leverages existing vision-language models (VLMs) to generate high-information-density scientific illustrations—such as model schematics and experimental workflows—by directly interacting with Microsoft Visio through the Model Context Protocol (MCP). Unlike conventional text-to-image models that produce non-editable raster outputs, VisPainter enables element-level control and iterative refinement via a collaborative pipeline: a Manager orchestrates tasks, a Designer proposes structured layouts, and a Toolbox executes atomic GUI operations in Visio while providing visual feedback. This closed-loop design supports true post-generation editability and aligns with real-world scientific illustration workflows.

To systematically evaluate such systems, the authors introduce VisBench, the first benchmark tailored for editable, high-density scientific diagrams. VisBench comprises 360 expert-annotated figures, sampled via a difficulty-balanced strategy based on element count. It assesses performance across seven dimensions: content fidelity, layout quality, visual perception, and interaction cost. A novel Dynamic Quality Score (DQS) integrates these metrics while penalizing inefficient generation processes.

Extensive experiments across nine VLMs validate the necessity of VisPainter’s modular design and demonstrate the robustness of VisBench. Ablation studies further underscore the benchmark’s reliability.

**Strengths:**

1. The paper presents a clear and well-structured formulation of the task, the multi-agent workflow, and the evaluation protocol, enabling readers to quickly grasp the core ideas and system design.
2. The proposed VisBench is a comprehensive and thoughtfully constructed benchmark. Its difficulty-balanced monthly cohort sampling strategy enhances the stability and fairness of model evaluation across time.
3. The authors conduct extensive experiments across multiple vision-language models and settings, thoroughly validating both the effectiveness of VisPainter and the reliability of VisBench through ablation studies and detailed capability analyses.

**Weaknesses:**

1. **Lack of direct comparison with alternative generation paradigms**: The paper claims that conventional text-to-image (T2I) models and code-based approaches (e.g., SVG/TikZ) are unsuitable for editable scientific illustration, but it does not provide side-by-side experimental evidence to substantiate this claim.

1.1 **Missing comparison with state-of-the-art unified vision-language models**: While the authors argue that raster-based T2I models cannot produce editable, high-fidelity scientific diagrams, they do not benchmark against recent models that combine strong generation and understanding capabilities—such as GPT-Image, or models like Nano-Banana, which are reportedly capable of generating structured visual content. Without such comparisons, the claimed limitation of T2I models remains anecdotal rather than empirically grounded.

1.2 **Insufficient justification for excluding SVG-based agent workflows**: The paper dismisses SVG as impractical due to its “write–compile–review” loop, yet SVG is inherently structured, editable, and supports incremental construction—properties well-aligned with agent-based iterative refinement. In principle, the VisPainter framework could be adapted to output SVG by replacing the Visio Toolbox with an SVG renderer and parser. Moreover, recent works, such as OmniSVG [1], demonstrate that the complex generation of SVG is possible. The authors should implement and evaluate an SVG variant of their pipeline to empirically justify the choice of Visio, especially since scientific figures often map naturally to vector primitives that SVG expresses natively.

2. **Limited model coverage in benchmarking**: Although VisBench evaluates nine vision-language models, the selection is heavily skewed toward proprietary, closed-API systems (e.g., GPT-4o, GPT-5, Claude-Opus-4, Gemini-2.5-Pro). Only three open-weight models are included (Qwen-VL-Max, Qwen2.5-VL-72B, Llama-4-Maverick), and even among these, key open-source contenders—such as InternVL, LLaVA-NeXT, or Yi-VL—are absent. Given that VisBench aims to serve as a community benchmark, broader inclusion of open models would strengthen its utility for reproducible and accessible research.

[1] Y. Yang, W. Cheng, S. Chen, X. Zeng, F. Yin, J. Zhang, L. Wang, G. Yu, X. Ma, and Y.-G. Jiang, “OmniSVG: A Unified Scalable Vector Graphics Generation Model,” arXiv preprint arXiv:2504.06263, 2025.

**Questions:**

1. The paper focuses exclusively on high-information-density scientific diagrams. However, VisPainter is built upon general-purpose vision-language models and operates through a pixel-level visual feedback loop with a vector graphics editor—components that are not inherently domain-specific. This raises a natural question: have the authors evaluated their framework on other types of visual content (e.g., technical sketches, educational illustrations, or architectural schematics)? Demonstrating generalization to related domains would strengthen the claim that the architecture could support even more complex diagrammatic tasks in the future.

The paper’s novelty appears limited, as the pipeline seems mainly a combination of off-the-shelf editing tools with multi-turn chats. While the Visio generation may be a key design choice, the authors do not provide solid ablation studies comparing it against prior generation approaches. Clarifying this—particularly in response to Weakness 1—is essential.

---

### Official Review · Reviewer_WL6o · 2025-11-01

**Soundness:** 2
**Presentation:** 2
**Contribution:** 2
**Rating:** 2
**Confidence:** 4

**Summary:**

The paper presents an MCP framework for evaluating (especially commercial) large language models for generating scientific illustrations that typically appear in academic venues like this conference. The authors have focused on two problems simultaneously: (1) translating instructions into vector diagrams and (2) evaluating the *communication quality* of the generated images. To address this issue, an *MCP tool over Microsoft Visio* called VisPainter made up of manager and designer agents. The tool is installed upon existing LLMs, and the results are benchmarked with 360 open source publications curated by the authors using various different scores.

**Strengths:**

- The problem setting of scientific illustration drawing is novel. Extendibility of this problem is not addressed well in the paper, so the problem appears to be quite narrow in scope. However, I believe this problem can be polished further into benchmarks for illustrative reasoning of language models.
- The introduction of MCP tools and their benchmarks in the major conference like this is quite a new approach, and I would like to mark this as a strength rather than weaknesses, despite its “high-level-ness” of the topic.
- The authors have provided diverse and various benchmark results.

**Weaknesses:**

Despite the novelty of this work, I find several unresolved issues within the presentation.

1. Although the authors have spared multiple pages to explain their philosophy behind the choices of quantitative scoring system, the complexity of the evaluation metrics are not fully justified. The central problem is that the authors have presented both the MCP framework and their evaluation metrics at the same time. They require in-depth justification to ensure that the performance measures are not biased, for example, by naming previous works that rely on the same measures, or by proving why each metric has to be used. Currently, such justification is not sufficient enough.
2. If the provided framework and the benchmark to be regarded as a general, reliable protocol, they should not be relying only on a specific commercial product of Microsoft Visio. A product-biased approach may be adequate for the technical report to aid specific user community. However, I believe such bias is not suitable for academic venues. Since the authors already encapsulated the toolbox with an MCP layer, we may expect multiple different commercial/open-source drawing software (e.g., Illustrator, TikZ, Inkscape, GIMP, Powerpoint, Visio, etc.) being used within the same framework. Such extension will be resolving this fundamental bias of this project.
3. The ablation study in Table 3 does not seem to show a reasonable trend, raising questions on the choice of quantitative measures used in the study. Moreover, the two blocks seem to be related to different models but this specification is not denoted in the table.
4. The criteria for the choice of 360 research papers for the benchmark is not fully justified. I do not oppose to *expert curation*; however, the transparency can be improved by displaying bad and good examples illustrating what type of figures are preferred in the benchmark. Also, the number of samples (360 + 120/season) seems to be small to evaluate LLMs for general capability. Furthermore, in line 1172-1185, the seasonal benchmark updating protocol seems to be more like a management manual of a product rather than an academic report on a firmly established methodology.

The most fundamental issues I take care of is number 1 and 2. Regarding the mentioned issues, I believe that the current manuscript is not ready for publication in this venue, yet. However, I am open to discussion with the author as well as with the other reviewers.

**Questions:**

I have specified what issues I take care most in the Weaknesses section. Please check the suggestions I made to resolve the issues.

These are questions that are minor enough that do not count in the final scores.

1. Text labels in Figure 2 and 4 are hard to read. Please reconsider rescaling them.

---

### Official Review · Reviewer_tRVW · 2025-11-01

**Soundness:** 2
**Presentation:** 3
**Contribution:** 2
**Rating:** 4
**Confidence:** 4

**Summary:**

This paper presents VisPainter, a multi-agent framework that generates editable scientific illustrations by orchestrating Microsoft Visio through GUI operations via the Model Context Protocol (MCP). The system employs a Manager-Designer-Toolbox architecture where the Manager handles task interpretation and tool dispatch, the Designer generates layout plans, and the Toolbox executes atomic drawing operations. The authors also introduce VisBench, a benchmark containing 360 high-information-density scientific diagrams with a seven-dimensional evaluation protocol assessing content fidelity, layout quality, visual perception, and interaction cost. Experiments across nine vision-language models demonstrate the framework's capabilities and validate the benchmark design through extensive ablation studies.

**Strengths:**

The paper addresses a genuine gap in scientific diagram generation by bridging the divide between raster-based generative models that lack editability and code-based approaches that impose cumbersome write-compile-review cycles. The originality lies in the GUI-level interaction paradigm, which enables direct manipulation of vector elements while maintaining full editability—a practical advantage over both existing approaches. The multi-agent architecture with explicit role separation is well-motivated through ablation studies that demonstrate catastrophic performance degradation when roles are merged, validating the design philosophy. The benchmark contribution is significant as VisBench represents the first systematic evaluation framework for high-information-density scientific schematics, going beyond simple charts or data plots to address complex workflows and model architectures. The difficulty-balanced sampling strategy using continuous element-count metrics rather than subjective categories is methodologically sound and addresses a real problem in benchmark design. The paper demonstrates strong empirical rigor through comprehensive ablations covering role configuration, step granularity, and description quality, with results showing that model rankings remain stable across variations. The integration of interaction cost (step count) into the Dynamic Quality Score represents a valuable perspective shift from purely output-focused metrics to process efficiency. The presentation is generally clear with well-structured sections, extensive appendices providing implementation details, and transparent discussion of limitations including speed constraints and platform dependency.

**Weaknesses:**

The computational efficiency represents a critical practical limitation, with complex diagrams requiring several tens of minutes to generate—orders of magnitude slower than single-pass diffusion models. This overhead severely restricts the framework's utility for rapid iteration or large-scale deployment. The tight coupling to Microsoft Visio fundamentally limits the work's impact and generalizability. While the authors justify this choice through development cost arguments, the 60:81 usage ratio they cite actually suggests PowerPoint has broader adoption, and the claim that both require similar development effort undermines the Visio-specific choice. The evaluation methodology, while extensive, contains several arbitrary components that weaken its rigor. The DQS formula (Equation 7) introduces hyperparameters K and r that are dataset-dependent and lack theoretical justification for their specific functional form. Similarly, the blank-space metric's 2β denominator (Equation 3) and alignment variance scaling by 10^4 (Equation 4) appear ad-hoc. The validation of the grid-based blank-space estimation uses only five images with 20 runs each—insufficient to establish robust statistical properties. The benchmark itself has limitations in scope and validation. With only 360 diagrams and monthly cohorts of 30, the sample sizes for drawing firm conclusions about model capabilities are modest. More critically, the paper lacks human performance baselines, making it impossible to contextualize the reported scores or understand how far models are from human-level capability. The absence of statistical significance testing, confidence intervals, or error bars across all experimental results undermines the reliability of claimed performance differences. The paper fails to provide direct empirical comparisons with code-based generation methods (TikZ, SVG) on the same prompts, making it impossible to assess whether the GUI-based approach actually outperforms alternatives or merely represents a different trade-off. The annotation quality issue is acknowledged but not adequately addressed—while ablations show ranking stability, the absolute validity of the ground truth remains uncertain. Several presentation issues detract from the paper's quality, including a typo in the abstract ("Frist" instead of "First"), inconsistent notation in some formulas, and figure quality that cannot be fully assessed from the text alone. The reproducibility is currently compromised as code and data release is conditional on acceptance rather than being available for review. Finally, the paper lacks substantive error analysis or failure case studies that would provide insights into fundamental limitations of the approach versus implementation-specific issues.

**Questions:**

First, can you provide a direct empirical comparison between VisPainter and code-based generation methods (AutomaTikZ, StarVector, OmniSVG) on identical prompts from VisBench? This would clarify whether the GUI-based approach offers genuine advantages or simply different trade-offs. Second, what is the sensitivity of the DQS metric to the hyperparameters K and r? A systematic ablation varying these parameters would help establish whether your model rankings are robust or artifact of specific parameter choices. Third, can you provide detailed computational cost analysis including wall-clock time, number of API calls, total tokens consumed, and monetary cost per diagram? This would enable practical assessment of the framework's feasibility. Fourth, while you show that rankings are stable across description quality variations, how do absolute scores change, and at what point does annotation quality become insufficient for meaningful evaluation? Fifth, what is the human performance ceiling on VisBench? Without this baseline, it is impossible to interpret whether scores of 0.6-0.8 represent near-human performance or substantial room for improvement. Sixth, can you provide a systematic failure mode analysis categorizing common error types (e.g., element omission, misalignment, incorrect connections) and their frequencies across models? This would offer actionable insights beyond aggregate scores. Seventh, the step-granularity ablation shows collapse at n≥8, but what is the underlying cause—context length limits, planning horizon, or something else? Understanding this mechanism would inform better system design. Eighth, your blank-space validation uses only five images—can you extend this to the full benchmark to establish robust statistical properties? Ninth, how does the framework handle diagrams that inherently require iterative refinement versus those that can be generated in a single pass? Is there a way to predict or optimize for this? Finally, given that Visio dependency is acknowledged as a limitation, what is the concrete plan and timeline for generalizing to other platforms, and what technical barriers exist?

---

### Official Review · Reviewer_56WJ · 2025-11-02

**Soundness:** 2
**Presentation:** 3
**Contribution:** 2
**Rating:** 4
**Confidence:** 3

**Summary:**

This paper introduces VisPainter, a multi-agent framework for generating editable scientific illustrations using Microsoft Visio via the Model Context Protocol (MCP). ​The framework includes three agents: Manager, Designer, and Toolbox. The authors also propose VisBench, a benchmark for evaluating high-information-density scientific diagrams using a seven-dimensional metrics.​ Extensive experiments validate the framework and benchmark, demonstrating the effectiveness in producing simple scientific illustrations.

**Strengths:**

- an agentic system for illustration generation with tool calling
- a new benchmark for science illustration generation with metrics
- provide controllability and edibility for output design

**Weaknesses:**

- agent system with prompts on LLM without much innnovation
- the generation results are very simple flow charts, yet the results is not great
- the benchmark size is limited, with only a few hundred annotated examples

**Questions:**

- comment how to generate more complex scientific illustrations, such as the Fig1 in the paper. what is lacking to achieve this complexity?
- compare with pixel image generation results, show the pros and cons compared with vector based results.

---

### Meta-Review · Area_Chair_e5n5 · 2026-01-02

**Summary:**

Reviewers raise the following concerns:
1. Reviewer 56WJ and Reviewer GfaX concerns about the lack of novelty for this paper.
2. Many reviewers share concerns about the experiment results. For example, Reviewer 56WJ noted that "results is not great". Reviewer tRVW concerns about experiment setup and Reviewer GfaX points out the lack of comparisons to proper baselines.
3. Reviewer WL6o pointed out that the paper didn't provide proper justification about the benchmark.

**Reviewer Concerns:**

I found no response from the authors.

**Reviewer Scores:**

I wouldn't expect the reviewers to change their scores given the lack of responses.

---

### Decision · Program_Chairs · 2026-01-26

Reject